# The tumor suppressor microRNA let-7 inhibits human LINE-1 retrotransposition

Pablo Tristán-Ramos [1,2], Alejandro Rubio-Roldan [1,5], Guillermo Peris [1,3,5], Laura Sánchez [1,5], Suyapa Amador-Cubero[1], Sebastien Viollet[4], Gael Cristofari [4] & Sara R. Heras [1,2 ✉]

Nearly half of the human genome is made of transposable elements (TEs) whose activity continues to impact its structure and function. Among them, Long INterspersed Element class 1 (LINE-1 or L1) elements are the only autonomously active TEs in humans. L1s are expressed and mobilized in different cancers, generating mutagenic insertions that could affect tumor malignancy. Tumor suppressor microRNAs are ~22nt RNAs that post-transcriptionally regulate oncogene expression and are frequently downregulated in cancer. Here we explore whether they also influence L1 mobilization. We show that downregulation of let-7 correlates with accumulation of L1 insertions in human lung cancer. Furthermore, we demonstrate that let-7 binds to the L1 mRNA and impairs the translation of the second L1-encoded protein, ORF2p, reducing its mobilization. Overall, our data reveals that let-7, one of the most relevant microRNAs, maintains somatic genome integrity by restricting L1 retrotransposition.

[1] GENYO, Centre for Genomics and Oncological Research: Pfizer/University of Granada/Andalusian Regional Government. PTS Granada, Av. de la Ilustración, 114, 18016 Granada, Spain. [2] Dept. Biochemistry and Molecular Biology II, Faculty of Pharmacy, University of Granada, Campus Universitario de Cartuja, 18071 Granada, Spain. [3] Dept. of Computer Languages and Systems, Universitat Jaume I, Castellón de la Plana 12071, Spain. [4] Université Côte d'Azur, CNRS, INSERM, IRCAN, Nice, France. [5] These authors contributed equally: Alejandro Rubio-Roldan, Guillermo Peris, Laura Sánchez. ✉email: sara.rodriguez@genyo.es

Transposable elements (TEs) account for nearly half of the human genome[1]. However, the only TE that remains autonomously active nowadays is a non-Long Terminal Repeat (non-LTR) retrotransposon known as Long INterspersed Element class 1 (LINE-1 or L1), whose mobilization continues to impact our genome[2]. LINE-1s comprise >20% of our DNA[1] but only about 80–100 of the ~500,000 L1 copies present in the average human genome are full-length elements that retain the ability to mobilize and are thus called Retrotransposition-Competent L1s (RC-L1s)[3]. RC-L1s belong to the human-specific L1Hs subfamily, are 6 kb long and encode two proteins (L1-ORF1p and L1-ORF2p) that are indispensable for retrotransposition[4]. However, ORF2p is expressed at a significantly lower level than ORF1p[5,6], and these differences are thought to be controlled at the level of translation[7]. L1-ORF1p is a 40 kDa RNA binding protein with nucleic acid chaperone activity[8,9], whereas L1-ORF2p is a 150 kDa protein with Endonuclease (EN) and Reverse Transcriptase (RT) activities[10,11]. RC-L1s mobilize by a "copy-and-paste" mechanism, involving reverse transcription of an RNA intermediate and insertion of its cDNA copy at a new site in the genome (reviewed in[2]). Briefly, retrotransposition starts with the transcription of a full-length RC-L1 bicistronic mRNA, which is exported to the cytoplasm and translated, giving rise to L1-ORF1p and L1-ORF2p that bind preferentially to the same L1 mRNA to form a ribonucleoparticle (RNP)[12]. The RNP gains access to the nucleus where retrotransposition occurs by a mechanism known as Target Primed Reverse Transcription (TPRT)[13,14]. During TPRT, the endonuclease activity of L1-ORF2p nicks the genomic DNA, and its reverse transcriptase activity uses the L1 mRNA as a template to generate a new copy of the element in a different genomic location. L1 can target all regions of the genome, but integration is locally dictated by the presence of a consensus sequence 5′-A/TTTT-3′, which is recognized by L1 endonuclease activity and allows annealing of L1 mRNA poly(A) to the target DNA[15,16]. Other non-autonomous retrotransposons such as Alu and SINE-R/VNTR/Alu (SVA) may hijack the L1-encoded proteins and be mobilized in trans[17,18]. Furthermore, L1-encoded proteins can sporadically generate pseudogenes using cellular mRNAs as templates[19].

TEs can affect genome stability in several ways, including the accumulation of insertions and rearrangements[2,20]. Genomic alterations caused by L1 activity have resulted in several human disorders[21]. Among these alterations, new L1 insertions can disrupt a gene unit, induce changes in splicing patterns, or interfere with transcription (reviewed in refs. [2] and[21]). Remarkably, new L1 insertions accumulate not only during early embryogenesis and in the germline, being transmitted to the next generation, but also in cancer cells, which are characterized by genome instability[22–33] (thoroughly reviewed recently in refs. [34,35]). In fact, L1s are highly expressed and mobilized in a wide range of human epithelial cancers[23,24], and high levels of L1 mobilization are found in lung and colorectal cancers[23,29]. Interestingly, several reports have shown that somatic L1 insertions can drive tumorigenesis and may even have initiated the tumor in normal cells[23,25,27,28,36]. Transcriptional control through methylation of the L1 promoter is one of the main defence mechanisms against L1 activity[37,38], and it has been demonstrated that hypomethylation of specific RC-L1s is associated with retrotransposition in early tumorigenesis[27,29]. However, additional post-transcriptional mechanisms that silence and reactivate L1 in somatic normal and tumor cells are not completely understood yet.

MicroRNAs (miRNAs) are small RNAs that are loaded into the Argonaute (AGO) proteins to form the RNA-Induced Silencing Complex (RISC) and post-transcriptionally repress gene expression[39,40]. Hundreds of *bona fide* miRNAs exist in humans and each of them is predicted to target many mRNAs[41].

Therefore, miRNAs could be influencing essentially all human developmental, physiological, and pathological processes[39]. In particular, overall miRNA dysregulation has been described in cancer[42,43]. Interestingly, it was previously shown that mouse embryonic stem cells (mESCs) lacking mature miRNAs (DGCR8 or Dicer knockout) accumulate LINE-1 mRNA[44–46]. Whereas the increase in LINE-1 mRNA levels in the absence of DGCR8 was attributed to reduced noncanonical functions of the Microprocessor, which cleaves stem-loops present in L1 elements[45], it remains possible that miRNAs regulate L1 expression levels. Consistently, a previous study reported that miR-128 represses engineered L1 retrotransposition in cultured cells[47]. Thus, we hypothesized that some miRNAs could control L1 retrotransposition and that their misexpression in tumors could contribute to increased LINE-1 mobilization.

To test this possibility, we first analyze whole genome sequencing data from a panel of human lung tumor/normal pairs and miRNA expression data from the same tumor samples. Notably, we find that samples containing tumor-specific L1 insertions express reduced levels of several members of the tumor suppressor miRNA let-7 family, suggesting that this miRNA could influence retrotransposition in vivo. Indeed, we further demonstrate that let-7 binds directly to the L1-mRNA impairing L1-ORF2p translation, and reduces L1 retrotransposition in cultured tumor cells. Altogether, our results uncover a role for let-7 in maintaining genome integrity and provide mechanistic insight into how downregulation of let-7 miRNAs in tumors may unleash L1 activity, causing genome instability and driving tumor genome evolution.

## Results

**miRNA levels correlate with L1 activity in lung cancer.** To identify potential miRNAs whose deregulation could produce a change in L1 retrotransposition in epithelial tumors, we focused on Non-Small Cell Lung Cancer samples (NSCLC) from The Cancer Genome Atlas (TCGA), as endogenous L1s are known to retrotranspose efficiently in this tumor type[23,29,33]. We selected all the samples (45 patients) for which whole-genome sequencing data from tumor and matched normal lung tissue, together with tumor miRNA-seq data, were available. We computationally identified tumor-specific somatic L1 retrotransposon insertions from whole-genome sequencing data using the MELT (Mobile Element Locator Tool) software[48]. Briefly, MELT detects Mobile Element Insertions (MEIs) by searching for discordant reads pairs and split reads that are enriched at genome positions containing new, non-referenced insertions[48]. First, to rule out possible biases produced by different coverage or quality of sample pairs, we analyzed the polymorphic germline L1 insertions identified by MELT both in tumor and normal tissue. We excluded four samples where this common reference polymorphic calls number was abruptly reduced under 10% after filtering. In the 41 selected samples, the number of polymorphic L1 insertions found in tumor/normal DNA pairs was similar and at least 63% of them were common to both DNAs (Supplementary Table 1). After exclusion of polymorphic L1s[49], we detected 413 putative de novo L1 insertions specific to cancer samples, which were absent in matched normal DNA from the same patient (Supplementary Table 1 and Supplementary Data 1). The low number of putative de novo insertions found in normal tissue but not in tumor tissue (3 in the 41 samples), expected to be zero, confirmed the specificity of the method. Consistent with previous studies, 409 of the 413 tumor-specific de novo L1 insertions identified here occurred in intronic and intergenic regions (Supplementary Data 1), likely representing passenger mutations[34,35].

To evaluate a possible correlation between L1 retrotransposition in lung cancer and miRNA expression, tumor samples were

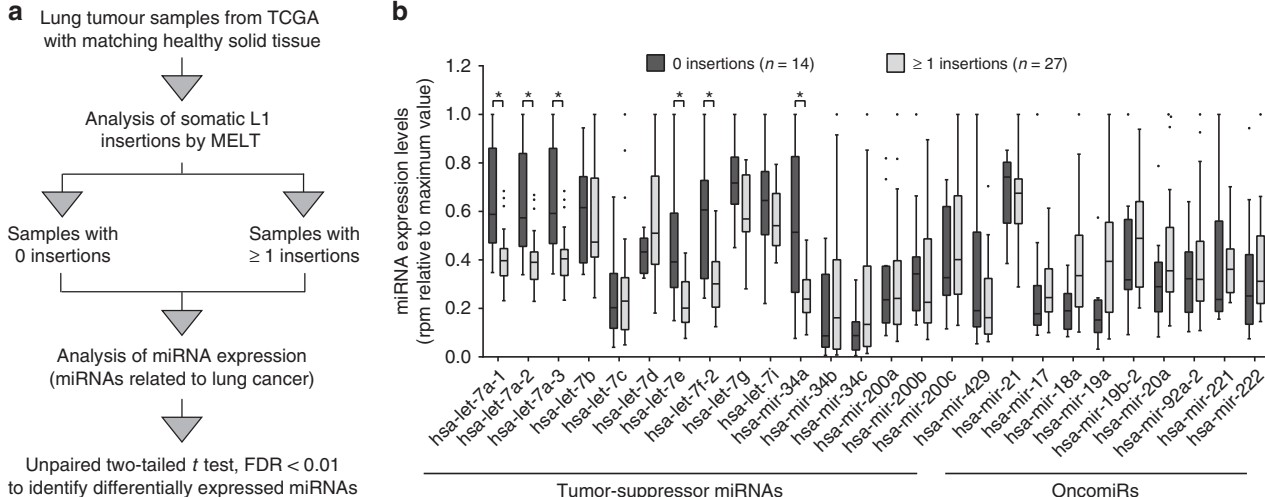

**Fig. 1 Let-7 and miR-34a are downregulated in lung tumor samples with tumor-specific L1 insertions. a** Schematic representation of the bioinformatic analysis used to identify differentially expressed miRNAs in lung cancer samples with or without tumor-specific L1 insertions. **b** A graph plot representing the expression levels of miRNAs previously associated with lung cancer[50] in lung tumor samples without (dark gray, $N = 14$) and with (light gray, $N = 27$) tumor-specific L1 insertions identified by MELT. Differentially expressed miRNAs are marked with * and were identified applying an unpaired two-tailed $t$ test adjusted by FDR < 0.01. To enable the representation of all miRNAs in one graph, expression in reads per million (rpm) was relative to the maximum value of each miRNA in each case. Source data are provided as a Source Data file. Whiskers were calculated using the Tukey method. Individual black dots represent outliers. Boxes extend from 25th to 75th percentiles, and lines in the middle of the boxes represent the median.

divided into two groups based on the presence (≥1) or absence (0) of tumor-specific L1 insertions (Fig. 1a). Using available miRNA-seq data across these samples in TCGA, we analyzed the expression of 26 miRNAs that have been previously associated with the development and/or progression of lung cancer, such as the let-7 family, the miR-34 family, or the miR-17-92 cluster[50] (Fig. 1b). Interestingly, we found that several members of the tumor suppressor let-7 family (let-7a, let-7e, and let-7f) were significantly downregulated in the samples with ≥1 tumor-specific L1 insertions upon multiple $t$ testing adjusted with False Discovery Rate (FDR) < 0.01 (Fig. 1b and Supplementary Table 2). This correlation was also found for let-7a and let-7f using a different statistical analysis (Rank-sum test, Supplementary Table 3). Although all the members of the let-7 family have a similar mature sequence, and could potentially bind to the same RNA targets, their genomic location and timing of expression are markedly different[51]. Interestingly, reduced expression of let-7a and let-7f has been observed in lung cancer samples[52,53]. Additionally, miR-34a, another tumor suppressor miRNA[54], was also significantly reduced in samples with tumor-specific L1 insertions (Fig. 1b and Supplementary Table 2). As a control, analysis was repeated after L1 insertion counts were randomly reassigned to each sample. No significant correlation was found in any case (one example is shown in Supplementary Fig. 1a and Supplementary Table 4). Notably, the differential expression of let-7a, let-7e, let-7f, and miR-34a was also significant in a more restrictive analysis where all the miRNAs expressed in lung tumor samples (89 miRNAs) were considered (Supplementary Table 5). Thus, even though we cannot rule out a possible bias in the analysis due to sample variability and the limited number of cases available, these data suggest that let-7 and miR-34a might control the accumulation of new L1 insertions in human lung cancer samples. Next, we used SQuIRE (Software for Quantifying Interspersed Repeat Elements)[55] to quantify L1Hs expression in RNA-seq data from these tumor samples, available in TCGA. As expected, L1Hs RNA levels were significantly increased in samples with tumor-specific L1 insertions (Supplementary Fig. 1b). However, L1Hs expression negatively correlates with miR-34a but not with let-7 expression (Supplementary Fig. 1b).

To further corroborate our results, we analyzed the correlation between miRNA expression and the number of tumor-specific L1 insertions identified by Helman and collaborators in a group of 46 lung tumor samples using Transpo-seq framework[23] (13 of them were also included in the previous analysis using MELT). Remarkably, the expression levels of let-7 family members (let-7a and let-7e) and miR-34a were again significantly reduced in those tumors containing tumor-specific L1 insertions when the 26 miRNAs related to lung cancer were analyzed (Supplementary Fig. 1c and Supplementary Table 6) as well as when all the miRNAs expressed in lung were included (Supplementary Table 7). Notably, the same analysis with the number of insertions randomly reassigned to each sample did not show any significant correlation with miRNA expression (Supplementary Fig. 1d and Supplementary Table 8). Lastly, the same analysis was performed using 36 breast cancer samples which contain a notably smaller number of tumor-specific L1 insertions per sample as determined by Transpo-Seq[23]. No significant correlation was found for any of the 26 miRNAs related to lung cancer (Supplementary Fig. 1e and Supplementary Table 9) suggesting that the contribution of let-7 and mir-34a to L1 mobilization could be specific to some tumor types.

Overall, these results suggest that a downregulation of let-7 and/or miR-34 expression can influence the accumulation of tumor-specific L1 insertions in lung cancer.

**Let-7 regulates human LINE-1 retrotransposition in vitro.** To investigate whether there is a causal relationship between the variation in let-7 and miR-34 expression levels and the accumulation of L1 insertions in tumors, we tested the effect of these miRNAs on L1 mobilization using the sRNA/L1 retrotransposition assay, recently developed in our lab[56]. This protocol combines the previously described cell culture-based LINE-1 retrotransposition reporter assay (reviewed in[2]) with miRNA mimics or inhibitors. Briefly, in this assay, cells are transfected with a plasmid containing an RC-L1 tagged with a reporter cassette (Fig. 2a). This cassette consists of a reporter gene (REP) in antisense orientation relative to the L1, equipped with its own promoter and polyadenylation signal, but interrupted by an

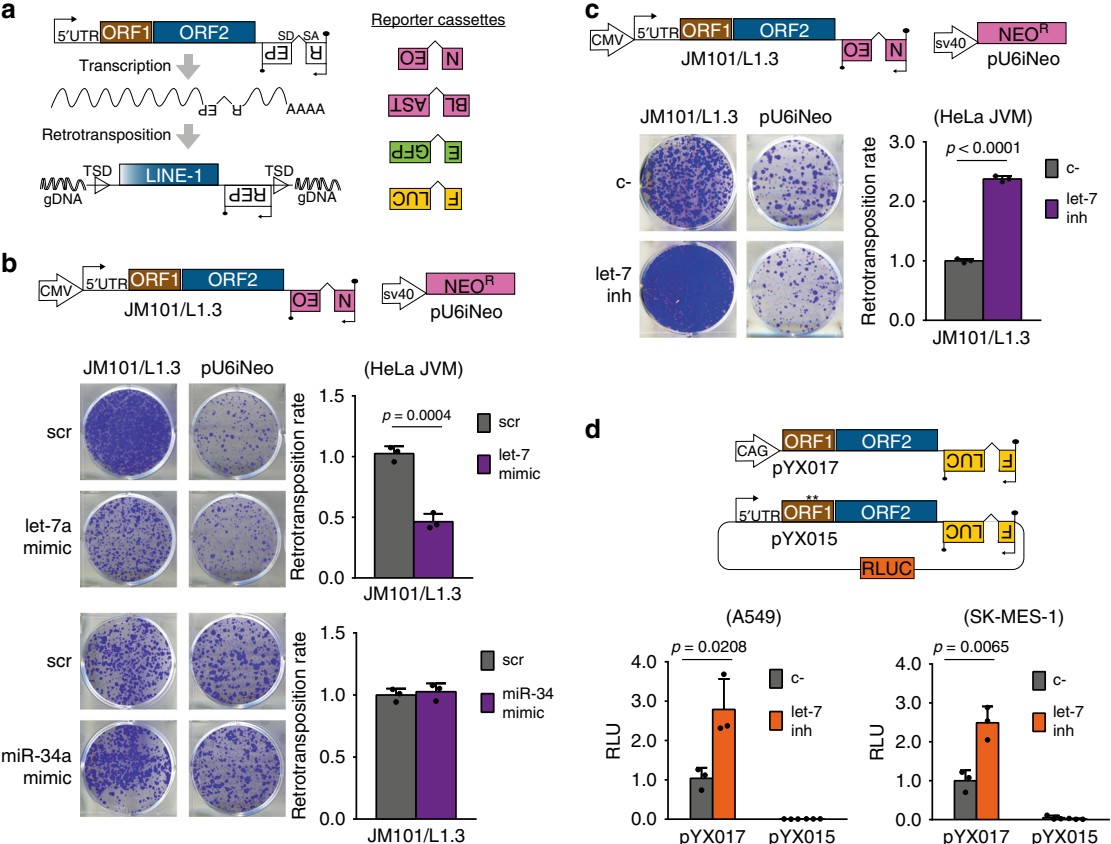

**Fig. 2 Let-7 regulates engineered human LINE-1 retrotransposition. a** Left panel: rationale of the retrotransposition assay in cultured cells. From left to right: transcription start site in the 5′UTR (black arrow), the two L1 open reading frames ORF1 (brown rectangle) and ORF2 (blue rectangle), the antisense-oriented reporter cassette (white rectangles, backward REP) interrupted by an intron, and the reporter gene promoter (inverted black arrow). Black lollipops represent poly(A) signals. TSD: Target Site Duplications. SD: Splicing Donor. SA: Splicing Acceptor. Right panel: reporter cassettes used in this study: neomycin (NEO or *mneoI*) and blasticidin (BLAST or *mblastI*) resistance, enhanced green fluorescent protein (EGFP or *megfpI*), and firefly luciferase (FLUC or *mflucI*). **b, c** Structures of pJM101/L1.3 and pU6iNeo constructs are shown. **b** HeLa cells were co-transfected with one of the plasmids and let-7a/miR-34a mimic and their control (scr). **c** HeLa cells were co-transfected with one of the plasmids and let-7 inhibitor and their control (c-). In (**b-c**), a representative well of three replicate is shown. Quantification of each experiment is shown at the right as average of three replicates ± s.d. **d** Structures of pYX017 and pYX015 are shown. CAG: Chicken Actin Globin promoter. The asterisk symbol in pYX015 indicates the two point-mutations in L1-ORF1p that abolish retrotransposition. Lung cancer cell lines A549 and SK-MES-1 were co-transfected with pYX017 or pYX015 and let-7 inhibitor or its control (c-). Luciferase activity was measured 96 h post-transfection, and Firefly luciferase signal was normalized to Renilla luciferase to correct for differences in transfection efficiency or cell survival. In both cases, averages of three replicates ± s.d. are shown. RLU: Relative Luminescence Units. In (**b-d**), an unpaired two-tailed *t* test was used to calculate *p* value, the statistically significant exact *p* values are shown. Source data are provided as a Source Data file.

intron located in the same transcriptional orientation as the L1. Thus, a functional reporter can only be produced after a successful round of retrotransposition (Fig. 2a). For this assay, we used cultured HeLa cells which express high levels of let-7a and almost undetectable levels of miR-34a as analyzed by RT-qPCR (Supplementary Fig. 2a). We next analyzed L1 activity upon overexpressing let-7a and miR-34a, using transfected synthetic miRNA mimics and a neomycin-resistance based retrotransposition assay (using plasmid JM101/L1.3, Fig. 2b and "Methods"). As a control, we performed a clonability assay co-transfecting the miRNA mimics with a plasmid encoding a constitutively-expressed neomycin-resistance gene (pU6iNeo) to rule out possible effects of miRNA overexpression on cell growth (Fig. 2b). In agreement with the above observation in lung tumor samples, we reproducibly detected a significant decrease in L1 retrotransposition upon overexpression of let-7a in HeLa cells without affecting the clonability of the cells (Fig. 2b, top panel). Furthermore, as expected, overexpression of different let-7 family members also reduced L1 retrotransposition (e.g. let-7b in Supplementary Fig. 2b). Strikingly, overexpression of miR-34 did not

affect L1 mobilization or cell clonability in this assay (Fig. 2b, bottom panel). Similarly, let-7 overexpression inhibits L1 retrotransposition in HEK293T cells, which express lower endogenous levels of this miRNA as compared to HeLa cells (Supplementary Fig. 2a). We used a dual-luciferase reporter vector containing a different RC-L1, L1RP (pYX014, Supplementary Fig. 2c). This plasmid uses Firefly luciferase as retrotransposition indicator and encodes a Renilla luciferase in the backbone to normalize for transfection efficiency (Supplementary Fig. 2c). Notably, we observed a consistent decrease in L1 retrotransposition upon co-transfection of the let-7 mimic in HEK293T cells (measured as the relative luminescence ratio (L1-Fluc/Rluc)) (Supplementary Fig. 2c). As expected, an inactive L1RP containing two missense mutations in the ORF1-encoded protein did not show luciferase activity (plasmid pYX15, Supplementary Fig. 2c). Considering that miRNAs downregulate the expression of their targets, the decrease of L1 mobilization upon let-7 overexpression suggests that L1 mRNA could be a *bona fide* let-7 target. Conversely, miR-34 overexpression in HEK293T cells, where the endogenous levels are slightly higher than in HeLa cells (Supplementary Fig. 2a), led

to an increase in L1 retrotransposition using the dual-luciferase reporter vector pYX014 (Supplementary Fig. 2d). The different effects observed for miR-34 overexpression in HeLa (Fig. 2b) and HEK293T cells (Supplementary Fig. 2d) suggest a potential indirect and cell-type specific effect of miR-34 on L1 mobilization.

To further investigate the role of let-7 on the control of L1 mobilization, we performed another panel of cell culture-based retrotransposition assays using a hairpin inhibitor to decrease intracellular let-7 levels. Although the inhibitor used was designed against let-7a, it has been shown to cross-react with other members of the family[57]. Consistent with our previous results, we found that depletion of let-7 in HeLa cells led to a two-fold increase in L1 retrotransposition without affecting the clonability of the cells using the neomycin-resistance cassette described above (Fig. 2c). A similar increase in L1 retrotransposition was observed in HEK293T cells upon let-7 depletion using an EGFP-based reporter cassette and a different human RC-L1, LRE3 (plasmid 99-UB-LRE3, Supplementary Fig. 2e). Furthermore, we confirmed that let-7 knock-down increased L1 retrotransposition in HEK293T using the luciferase reporter vectors pYX014 and pYX017 (Supplementary Fig. 2f). While both contain the same active human L1, L1RP, in pYX014 it is transcribed from the native promoter in the 5′UTR whereas, in pYX017, it is highly transcribed from a CAG promoter.

Lastly, since our bioinformatic analysis showed an inverse correlation between let-7 expression and accumulation of L1 insertions in human lung tumor samples, we analyzed whether let-7 could regulate L1 retrotransposition in lung cancer cells. To do that, we performed the luciferase-based retrotransposition assay in two lung cancer cell lines with markedly different endogenous levels of let-7, A549, and SK-MES-1 (Supplementary Fig. 2g). Interestingly, we observed that, in both cell lines, depletion of let-7 increased L1 retrotransposition by 2.5 times on average (Fig. 2d). Altogether, these data indicate that let-7 negatively regulates human L1 mobilization in a variety of cancer cell lines.

**Let-7 binds directly to the coding sequence of L1 mRNA**. The aforementioned regulation could occur either by a direct interaction between let-7-guided RISC and L1 mRNAs, or by an indirect effect, since let-7 could be regulating any host factor involved in the multiple steps of the retrotransposition cycle[58] or in L1 control[59]. Since a direct effect would be sequence-dependent, we performed a neomycin-resistance based retrotransposition assays in HeLa cells using non-human active LINEs, that differ in sequence from the human L1 but use the same target-primed reverse transcription mechanism for mobilization. Briefly, we used mouse TGF21 (L1GF subfamily) and zebrafish L2-1 and L2-2 (L2 clade). Structures of the different LINEs are shown in the left panel of Fig. 3a, and constructs are described in the Methods section. Interestingly, we observed that only human L1 mobilization was significantly affected by either the inhibition (Fig. 3a) or the overexpression (Supplementary Fig. 3a) of let-7. These results suggested a direct, sequence-dependent interaction between let-7 and human L1 mRNA.

It is well established that miRNAs mostly bind their target mRNAs in their 3′UTRs[39], although 5′UTR and coding sequence (CDS) binding sites have been described and validated[60–62]. Thus, to find out where the putative let-7 binding site was located in L1 mRNA, we performed the same retrotransposition assays but using an engineered human RC-L1 (L1.3) lacking either the 5′or the 3′UTR (Fig. 3b and Supplementary Fig. 3b). Notably, the effect of let-7 depletion or overexpression in engineered L1 mobilization was not abolished or reduced by the absence of

either 5′ or 3′ UTR, suggesting that let-7 interacts with the CDS of human L1 mRNA (Fig. 3b and Supplementary Fig. 3b).

It has previously been described that L1-ORF1p often aggregates in cytoplasmic foci and colocalizes with L1 mRNA and AGO2 protein, the main component of the RISC complex[63,64]. We further analyzed whether let-7 guides the RISC complex to L1 mRNAs by RNA-Immunoprecipitation (RIP). For this, we used a human embryonic teratocarcinoma cell line (PA-1), characterized by high levels of endogenous LINE-1 mRNA and L1-ORF1p[65] and very low levels of let-7 miRNAs (Supplementary Fig. 2g). Briefly, we overexpressed FLAG-tagged AGO2, pulled it down, purified the endogenous bound RNAs, and analyzed them by RT-qPCR (Fig. 3c). We reasoned that if let-7 can bind L1 mRNA, let-7 overexpression should lead to an increase in the abundance of endogenous L1 mRNAs associated to AGO2. Strikingly, we observed an enrichment in the amount of L1 mRNA bound to AGO2 upon overexpression of let-7 resembling the behavior of HMGA2 mRNA (Fig. 3d), a well-known target of let-7[66]. In contrast, none of the negative controls used, GAPDH and actin mRNAs, were enriched in the immunoprecipitation (Fig. 3d). Thus, these data suggest that let-7 guides Argonaute proteins to L1 mRNA, and that this interaction occurs within the L1 CDS.

**A functional let-7 binding site is located in L1-ORF2**. We next set out to predict and validate putative let-7 binding sites within the CDS of the L1 mRNA. We used two different software available online: miRanda[67] and RNA22[68]. The best predicted binding site for let-7 family by each method was located in positions 2650-2671 (bs1) and 4596-4616 (bs2), respectively, in the consensus L1Hs sequence (top panel, Supplementary Fig. 4a). In order to validate them, five tandem copies of each binding site (bs) were cloned in the 3′UTR region of the Renilla luciferase (Rluc) gene in the psiCHECK-2 vector, which also encodes a Firefly luciferase (Fluc) gene to correct for transfection efficiency (left panel, Fig. 4b). As controls, an unrelated sequence of the same length and a sequence with perfect complementarity to let-7 were cloned (no bs and perfect bs, respectively). Those constructs were co-transfected with let-7 mimic in HEK293T cells. The reporter constructs containing the RNA22-predicted binding site (bs2) and the positive control (perfect bs), but not the one with the miRanda-predicted binding site (bs1) or the negative control (no bs), showed a reduction of the relative luciferase ratio (RLuc/FLuc) upon let-7 overexpression (bottom panel, Supplementary Fig. 4a). A deeper analysis of the residues in this region interacting with let-7 miRNA using RNAhybrids software[69] suggests that the functional "bs2" is located within the CDS of L1-ORF2 (position 4587-4610 in L1.3, a commonly used human RC-L1, accession code # L19088.1[70]), between the RT and Cysteine-rich domains of this protein (Fig. 4a, left panel). Importantly, it is predicted to form a duplex with let-7 miRNA consisting of seven Watson–Crick pairings at positions 3–9 followed by an adenine at the mRNA nucleotide corresponding to the first nucleotide position of the miRNA, resembling a previously described functional noncanonical binding site termed offset 7-mer (Fig. 4a)[71]. Altogether, these results suggest that this refined binding site, hereafter referred to as "bs2rh" is a *bona fide* let-7 binding site. To further validate this binding site ("bs2rh") we generated a mutant sequence ("bs2rhmut", see Fig. 4b, left panel). Mutations introduced in "bs2rh" are predicted to severely impede the duplex formation between L1 mRNA and let-7 (Supplementary Fig. 4b). Accordingly, the mutated sequence rescued the luciferase activity upon overexpression of let-7 (bs2rh vs bs2rhmut, Fig. 4b). To further corroborate the functionality of "bs2rh" in the context of retrotransposition, we generated an allele mutated

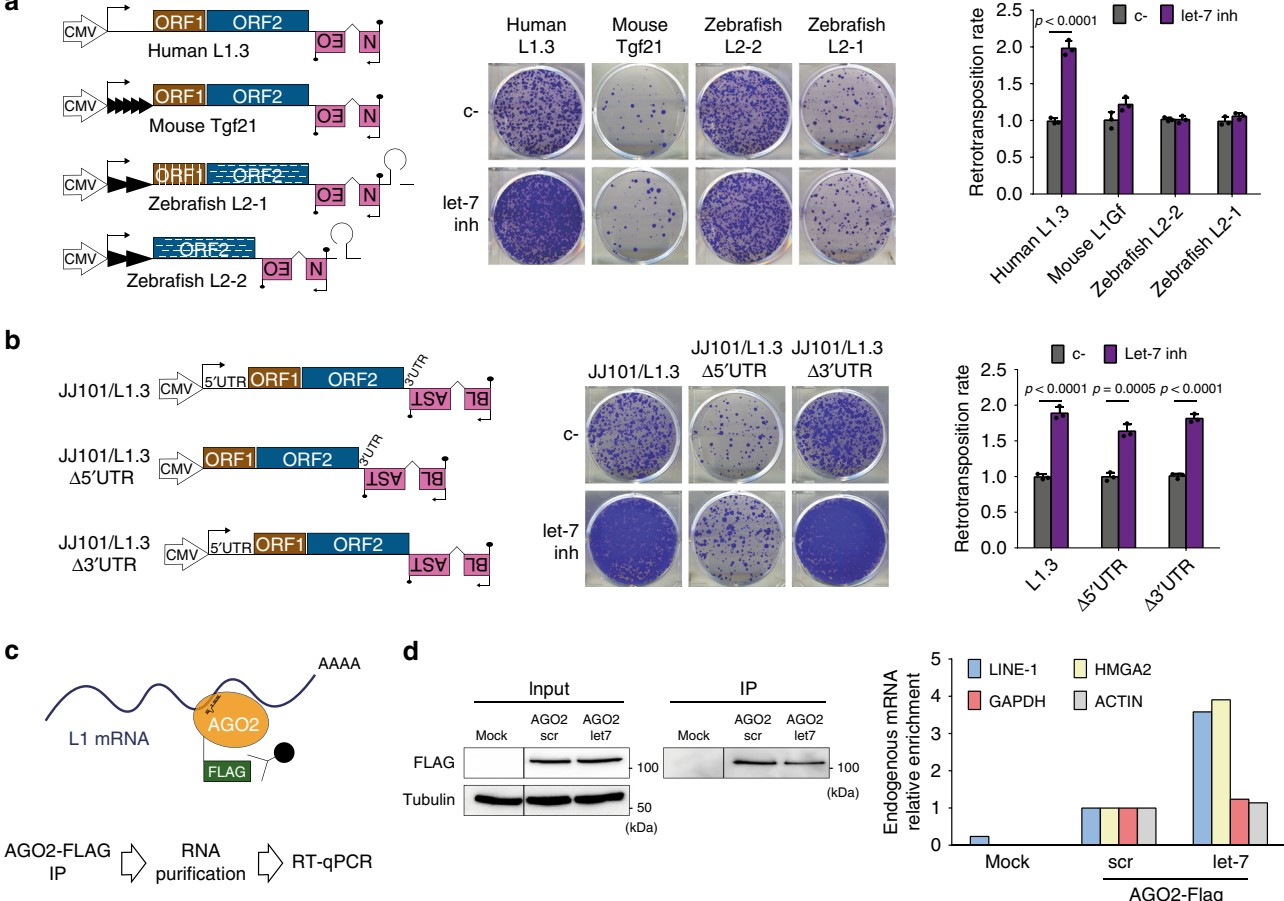

**Fig. 3 Let-7 binds directly to the coding sequence of L1 mRNA. a** Cell culture-based retrotransposition assay using *mneol* reporter cassette. HeLa cells were co-transfected with LINEs from different species and let-7 inhibitor or its control (c-). All constructs have an exogenous CMV promoter to normalized transcription. Black arrows represent transcription start sites. Gray triangles in mouse and zebrafish LINEs illustrate the presence of monomers in the 5′ UTR of these elements. Stem loop (gray) pictures the hairpin structures present in the 3′ UTR of the zebrafish LINE-2s, required for retrotransposition. White stripes are included to remark the differences in sequence of zebrafish L2-2 and L2-1 with respect to the human L1.3 and mouse L1G_F. A representative well is shown in the middle panel. Quantification is shown on the right as average ± s.d of three biological replicates. **b** Cell culture-based retrotransposition assay with blasticidin resistance cassette. HeLa cells were co-transfected with LINEs lacking either the 5′ or the 3′ UTR (structures shown in the left) and let-7 inhibitor or its control (c-). A representative well of three replicates is shown. Quantification is shown at the right as average of three replicates ± s.d. In (**a**–**b**), an unpaired two-tailed *t* test was used to calculate *p* value, and statistically significant exact *p* values are shown. Source data are provided as a Source Data file**. c** Scheme of RNA Immunoprecipitation (RIP) of AGO2-FLAG and RT-qPCR analysis of endogenous mRNA enrichment upon let-7 overexpression. Embryonic teratocarcinoma cells (PA-1) were co-transfected with a plasmid to overexpress AGO2-FLAG and let-7 mimic. AGO2-FLAG (orange circle with green flag) was immunoprecipitated with a FLAG antibody (black circle and lines), and the RNA bound to AGO2 (L1 mRNA is shown in blue) was purified and analyzed by RT-qPCR. Transfection with pBSKS (empty vector) was used as a negative control. **d** Real-time RT-qPCR analysis of endogenous L1 mRNA upon immunoprecipitation of AGO-2-FLAG of one representative experiment of three replicates. Left panel: loading controls are shown for input and IP. Right panel: mRNA relative enrichment upon let-7 overexpression: LINE-1 (blue), HMGA2 (yellow), GAPDH (red), and ACTIN (gray) are shown.

RC-L1 construct containing a mutated "bs2rh" site (we introduced the same mutation described above, construct JM101/bs2rhmut L1.3, Fig. 4c). Intriguingly, the validated "bs2rh" site is conserved through primate L1 evolution, being present in L1PA5 elements and containing a few mutations in older L1 subfamilies (Supplementary Fig. 4c). Accordingly, the introduction of the mutation contained in the "bs2rhmut" sequence, which entails an amino acid change (P to G) in L1-ORF2p, leads to a reduction in RC-L1 mobility (Fig. 4d, right side graph). We observed that "bs2rhmut" L1.3 retrotransposition was less affected by let-7 inhibition than wild-type L1.3 (Fig. 4d). Interestingly, this binding site is absent in zebrafish LINEs and relatively low conserved in mouse RC-L1s (Supplementary Fig. 4d) in agreement with the specific let-7 effect on human L1 retrotransposition showed above (Fig. 3a and Supplementary

Fig. 3a). However, the fact that mutating this binding site reduced but not abolished the effect of let-7 in L1 mobilization suggests that additional mechanisms mediated by let-7 may work to restrict human L1 retrotransposition. Overall, these results suggest that there is at least a functional let-7 binding site in the ORF2 region of human L1 mRNA.

**Let-7 impairs L1-ORF2p translation.** The above experiments identified a functional let-7 binding site in L1-ORF2p, and we next analyzed the functional consequences of let-7 binding to L1 mRNA. Since miRNAs can induce mRNA degradation[40], we analyzed the levels of endogenous L1 mRNAs upon let-7 over-expression in HEK293T cells by RT-qPCR. We found no changes in L1 mRNA levels at 24 and 48 hours after let-7 overexpression,

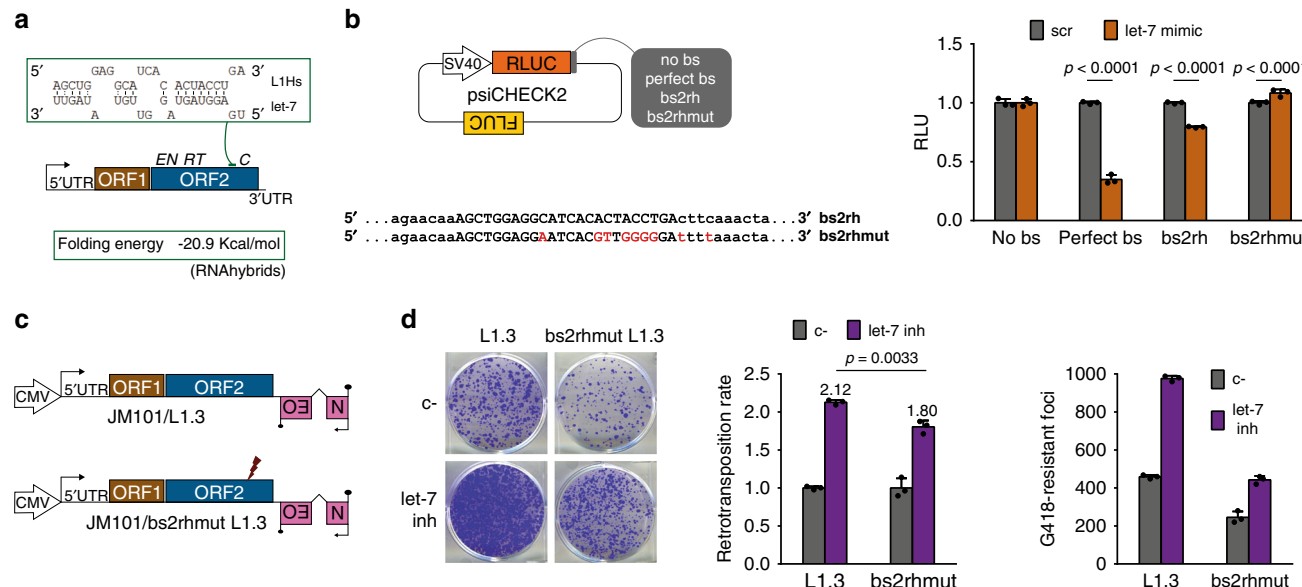

**Fig. 4 L1 mRNA contains a functional let-7 binding site located in L1-ORF2. a** RNAhybrid prediction of the best binding site for let-7 located in the region of L1Hs identified as "bs2" with RNA22. Base-pairing between this region and let-7b is shown (green rectangle). Localization of the putative binding site ("bs2rh") within L1 sequence is shown (green line). Structure of LINE-1 is shown: transcription start site (black arrow), 5′ untranslated region (UTR), ORF1, ORF2 with its three domains endonuclease (EN), reverse transcriptase (RT), and cysteine-rich (C), and 3′ UTR. Folding energy of the predicted binding site is shown below. **b** psiCHECK2 assay with "bsrh" and "bs2rhmut". Left panel: scheme of psiCHECK2 plasmid containing SV40 promoter, Renilla luciferase gene (RLuc, orange rectangle), different sequences cloned in the 3′UTR of the RLuc gene (gray rectangle), and Firefly luciferase gene (FLuc, brown rectangle). Comparison of the sequences cloned in psiCHECK2 as "bs2rh" and "bs2rhmut" is shown below, with the nucleotides interacting with let-7 in capital letters and the different nucleotides highlighted in red. Right panel: HEK293T cells were co-transfected with the different psiCHECK2 constructs and scr or let-7 mimic. Data are presented as mean values ± s.d. of three replicates. Source data are provided as a Source Data file. RLU: Relative Luminescence Units. Statistically significant exact *p* values are shown after applying an unpaired two-tailed *t* test. **c** Scheme of the wild type L1.3 and the binding site mutant "bs2rhmut L1.3" generated by site-directed mutagenesis. Red thunder indicates location of the mutated binding site. **d** HeLa cells were co-transfected with JM101/L1.3 or JM101/bs2rhmut L1.3 and let-7 inhibitor or its control (c-). A representative well of three replicates is shown. Quantification is shown as retrotransposition rate (relative to c-) and raw colony count, in both cases as average of three replicates ± s.d. Source data are provided as a Source Data file. Statistically significant exact *p* values are shown after applying an unpaired two-tailed *t* test.

whereas those of other canonical let-7 targets (DICER and HMGA2) were reduced (Fig. 5a). Similarly, L1 mRNA levels were not decreased upon let-7 overexpression (Supplementary Fig. 5a) or increased upon let-7 depletion (Supplementary Fig. 5b) when L1 was overexpressed in HEK293T cells. Thus, these data suggest that let-7 expression does not trigger L1 mRNA degradation.

The other main effect of miRNAs on their target mRNAs is interference with protein translation[72], so we analyzed the levels of endogenous L1-ORF1p upon modulation of let-7 levels in HEK293T cells. We found significant changes in HMGA2 but not in ORF1p expression upon let-7 overexpression (Fig. 5b) or depletion (Supplementary Fig. 5c). We corroborated this results in a stable HEK293T cell line that constitutively overexpresses a T7-tagged L1-ORF1p (from L1.3, and using a CMV promoter) (Supplementary Fig. 5d).

We next analyzed changes in L1-ORF2p levels. The translation of ORF2p occurs by a highly inefficient unconventional termination/reinitiation mechanism that, although could produce as few as one L1-ORF2p molecule per L1 mRNA[7], is enough to support efficient retrotransposition[4,7]. Consequently, it is technically challenging to detect endogenous L1-ORF2p. Thus, to study L1-ORF2p levels upon let-7 modulation, we generated a monocistronic construct expressing 3xFLAG-tagged ORF2p from a CMV promoter (L1-ORF2p from L1.3), pSA500. Strikingly, we observed an increase in ORF2p upon let-7 depletion and a decrease upon let-7 overexpression in HeLa cells (Fig. 5c) resembling the effect on DICER protein levels, a well-described target of let-7 with several 8-mer sites[60] (Fig. 5c). To rule out that differences in L1-ORF2p expression were due to different

transfection efficiencies, we took a fraction of each sample, extracted DNA, and quantified plasmid levels by qPCR using primers targeting the CMV promoter driving ORF2 expression or the EBNA-1 sequence in the plasmid backbone. We did not observe any significant differences in the amount of plasmid co-transfected with let-7 mimic (Supplementary Fig. 5e) or let-7 inhibitor (Supplementary Fig. 5f). Consistent with the data presented above, the difference at protein level neither correlates with changes in the levels of exogenous L1 ORF2-FLAG RNA (ORF2-F), as opposed to DICER whose mRNA is also reduced (bottom panel, Fig. 5c). These data suggest that the differences in ORF2p levels are not due to variations in transfection or mRNA accumulation but to an effect of let-7 on ORF2-F translation.

To understand whether let-7 mediated translational repression of ORF2p is due to the specific interactions with the offset 7-mer site or to its location within the CDS, we generated three variants of pSA500 in which we introduced different sequences in its 3′ UTR: a scrambled sequence ("scrb"), the binding site ("bs2rh") and a modified bs2 that contains a canonical 8-mer site for let-7 ("8mer") (Supplementary Fig. 5g). We co-transfected all these constructs in HeLa cells with let-7 mimic. First, by RT-qPCR we observed that similar levels of transfection (measuring constitutive EBNA expression from the plasmid backbone, Supplementary Fig. 5h) and let-7 overexpression (measuring the effect on endogenous DICER, Supplementary Fig. 5h) were achieved. The levels of ORF2 mRNA were not significantly affected in any case, although we observed a tendency towards a reduction on the RNA levels upon placement of the binding site ("bs2rh") or the modified 8-mer binding site ("8mer") in the

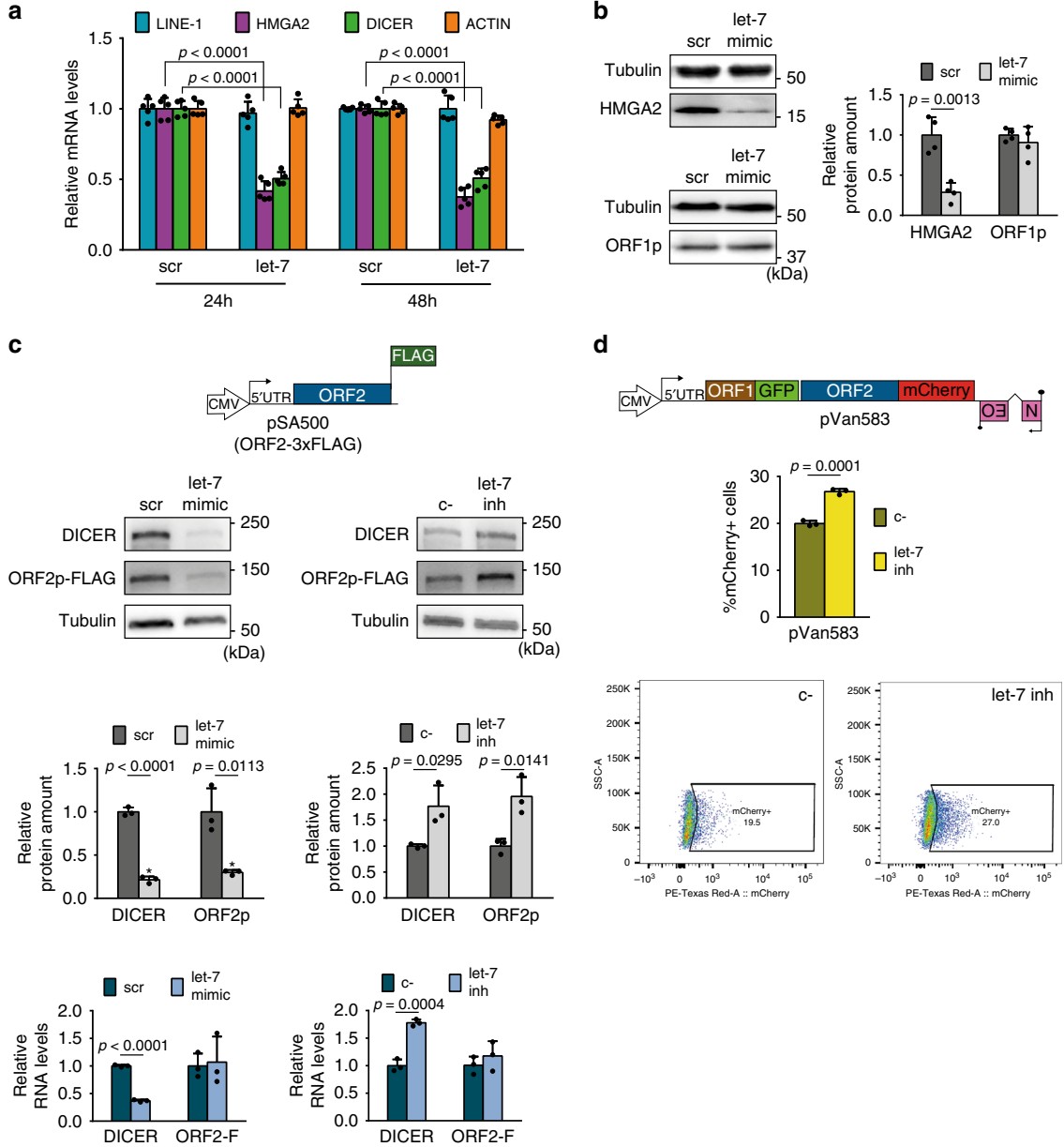

**Fig. 5 Let-7 impairs L1-ORF2p translation. a** RT-qPCR analysis of endogenous LINE-1 (blue bar), HMGA2 (purple bar), DICER (green bar), and ACTIN (orange bar) mRNAs upon let-7 overexpression in HEK293T. Cells were transfected with let-7 mimic or its control (scr), and RNA was extracted at 24 h or 48 h post-transfection. GAPDH was used to normalize. Data are presented as average ± s.d. of five biological replicates. **b** Western-blot analyses of endogenous L1-ORF1p and HMGA2 protein levels in HEK293T cells upon let-7 overexpression. Cells were transfected with let-7 mimic. Representative well and quantification of the western blot are shown. Data are presented as average ± s.d. of four replicates. **c** Western-blot analyses of L1-ORF2p-FLAG upon let-7 overexpression or depletion in HeLa cells. A scheme of construct pSA500 is shown. HeLa cells were co-transfected with pSA500 and let-7 mimic or inhibitor and their controls (scr or c- respectively). L1-ORF2p was detected using a FLAG antibody. DICER, a known let-7 target, was used as a positive control. Representative well and quantification of the western blot are shown. Below, RT-qPCR analyses of the levels of DICER and L1-ORF2-3xFLAG mRNA (ORF2-F) upon overexpression or depletion of let-7. GAPDH was used to normalize. Data are presented as average ± s.d of three replicates. **d** Flow cytometry quantification of L1-ORF2p-mCherry levels upon let-7 depletion in HeLa cells. The structure of construct pVan583 is shown. pVan583 is a derivative of JM101/L1.3 with L1-ORF1p fused to EGFP and L1-ORF2p fused to mCherry both at the C-terminus. HeLa cells were co-transfected with pVan583 and let-7 inhibitor or its control (c-), and fluorescence was measured by flow cytometry. Graph shows the percentage of mCherry+ cells in the EGFP+ (transfected) population. Data are presented as average ± s.d. of three replicates. A representative FACS histogram of three replicates in each condition is shown (the percentage of ORF1p-GFP positive cells expressing ORF2p-Cherry protein). In (**a**–**d**), an unpaired two-tailed *t* test was used to calculate *p* value, statistically significant exact *p* values are shown. Source data are provided as a Source Data file.

3′UTR (Supplementary Fig. 5h). Furthermore, western blot analysis showed that placement of "bs2rh" sequence in the 3′UTR of pSA500 slightly enhanced the reduction of ORF2-F protein upon let-7 overexpression (Supplementary Fig. 5i), an effect that was more prominent when the canonical site ("8mer")

was tested. In agreement with previous studies[71], these results suggest that the proficiency of "bs2rh", a noncanonical offset 7mer site, is weaker than that of a canonical let-7 binding site when they are located in 3′UTR. Moreover, we cannot rule out that the translational repression mediated by both binding sites

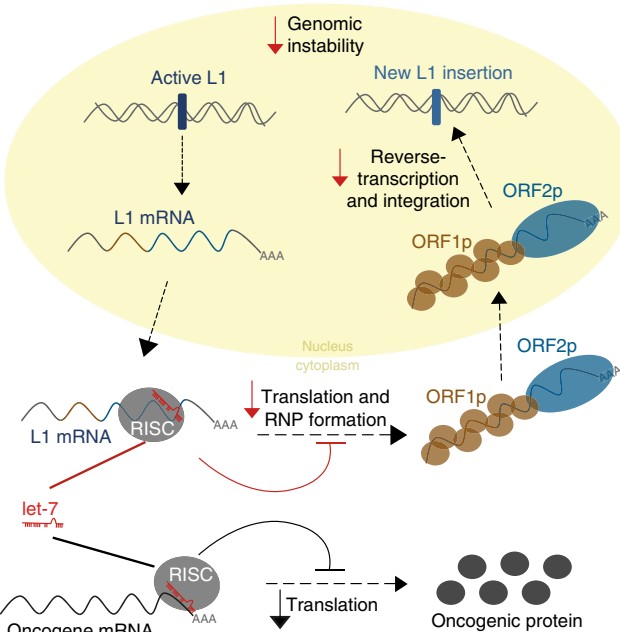

**Fig. 6 Model for the control of LINE-1 retrotransposition by the tumor suppressor miRNA let-7.** Besides the well-known regulation of oncogenes (lower part of the scheme in gray), we propose a novel tumor-suppressor role for let-7 miRNAs (upper part of the scheme in color). Once LINE-1 RNA is transcribed from an active L1 located in the genome and is exported into the cytoplasm, let-7 (drawn in red) binds and guides the RISC complex (gray circle) to the L1 mRNA (blue line). This binding leads to the inhibition of ORF2p translation (blue circle) and consequently, impairs the formation of the ribonucleoparticle (ORF1p (brown circle) and ORF2p (blue circle)). The reduction in ORF2p levels results in a decrease in the reverse transcriptase activity in the nucleus and the number of new L1 copies integrated in the genome, consequently, reduces the L1-associated genomic instability.

located in 3′UTR could be attributed to mRNA destabilization. Additionally, using site-directed mutagenesis we introduced two point-mutations in the ORF2 coding region to transform the offset 7-mer into a canonical let-7 8-mer site, generating pSA500-ORF2-8mer (Supplementary Fig. 5j). We co-transfected this construct in HeLa cells with let-7 mimic. Interestingly, the 8-mer site within the ORF does not affect the levels of mRNA (Supplementary Fig. 5k) and leads to a decrease in the protein level similar to that observed above for "bs2rh" (Supplementary Fig. 5l). Altogether, these results suggest that the translational repression mediated by "bs2rh" mainly depends on its location within the CDS, rather than on its noncanonical interaction.

To further characterize this effect, we next analyzed the impact of let-7 binding on L1-ORF2p translation in its natural context: a full-length bicistronic L1 RNA where L1-ORF2p is translated using the aforementioned termination/reinitiation mechanism[7]. In order to perform a more quantitative analysis, we combined the use of L1-encoded proteins with fluorescent tags and confocal microscopy or flow cytometry. Briefly, we generated a construct where L1-ORF1p and L1-ORF2p from a human L1.3 element were fused to EGFP and mCherry, respectively, at their C-terminus (plasmid pVan583, Fig. 5d). First, we confirmed that ORF1p-GFP was expressed from this construct (Supplementary Fig. 5m) and that this tagged L1 was able to retrotranspose, although the addition of both florescent tags reduced its activity to ~30% of its untagged counterpart, JM101/L1.3 (Supplementary Fig. 5n). Next, by confocal microscopy we observed a reduction of L1-ORF2p-mCherry but not of L1-ORF1p-EGFP levels upon

overexpression of let-7 in U2OS cells (Supplementary Fig. 5p). However, due to reduced transfection capacity of this construct (Supplementary Fig. 5o) and the inefficient translation of ORF2p-mCherry[7], we obtained an insufficient number of double positive cells to enable a quantitative analysis by microscopy. Therefore, we turned to a more sensitive and quantitative approach: flow cytometry. We co-transfected pVan583 with let-7 inhibitor in HeLa cells and analyzed EGFP⁺ cells (i.e. >3500 cells per sample). Notably, we found that depleting let-7 led to an increase in the number of mCherry⁺ cells in the EGFP⁺ population suggesting an increase in the synthesis of L1-ORF2p-mCherry (Fig. 5d).

Altogether, our results suggest that let-7 impairs L1-ORF2p translation, potentially altering the ratio between L1-ORF1p and L1-ORF2p, which we speculate could unbalance L1-RNP formation (Fig. 6).

## Discussion

Many studies have linked LINE-1 retrotransposons to cancer[22–29,31–33]. In particular, L1 insertions have been found to occur at high frequencies in lung cancer genomes[33]. L1 is also associated with genomic instability, since new insertions can potentially cause splicing alterations, exon disruptions, indel mutations, or large genomic rearrangements[2,20,36]. How these elements are silenced and derepressed in somatic human tissues, and how these processes impact tumorigenesis is an open question. DNA methylation of the L1 promoter is an important inhibitor of L1 activity[37]. In fact, a consistent correlation between the number of somatic L1 insertions in lung cancer and hypomethylation of L1 promoters has been shown, both at a global and at a locus specific level[29,38]. However, considering the high level of somatic L1 activity in some of these patients, it is tempting to speculate that RC-L1s might also escape post-transcriptional restriction mechanisms[44,73]. On the other hand, among all tumor suppressor miRNAs, reduced let-7 expression occurs most frequently in cancer and typically correlates with poor prognosis[74]. Functionally, it is well known that a decrease in let-7 miRNAs leads to overexpression of their oncogenic targets such as MYC, RAS, HMGA2 among others[51].

Here we describe a role for let-7 in controlling human L1 activity, which may contribute to its tumor suppressor function. First, we found a high frequency of retrotransposition in NSCLC cancer, consistent with previous reports[23,33]. We further showed that human tumor samples with somatic L1 insertions present reduced let-7 expression. Additionally, we demonstrated that the mobilization of full-length L1s in cultured cells can be negatively regulated by let-7 in a variety of cell lines including lung cancer cells.

It is worth noting that the expression of another tumor suppressor miRNA, miR-34a, is also reduced in lung tumors with L1 activity and correlates negatively with L1Hs RNA levels. However, we did not observe a consistent effect of the latter on L1 retrotransposition, under our experimental conditions. We speculate that mir-34 could indirectly regulate L1 mobilization, targeting a member of the epigenetic regulatory network controlling the expression of active L1s in our genome.

AGO proteins are the main effectors of miRNA-guided gene silencing[75]. Our AGO2 RNA-immunoprecipitation assay and the use of engineered retrotransposition constructs lacking 3′UTR or 5′UTR suggest that let-7 is actually guiding AGO2 to the human L1 mRNA, and that its binding occurs in L1 CDS. In fact, we have demonstrated that ORF2 contains a noncanonical offset 7-mer let-7 binding site previously described as functional for miRNA targeting[71]. Although it is widely accepted that most binding sites are located in the 3′UTR of target mRNAs[39], functional CDS binding sites for different miRNAs, including let-7, have also been described[60,61,76].

Furthermore, we demonstrate that mutations in this binding site reduce, but not abolish, the effect of let-7 modulation on human L1 mobility. These results suggest that additional let-7 binding sites may exist within the CDS of human L1 mRNA, or that let-7 might have redundant indirect effects. However, we failed to validate functional let-7 binding sites in a different L1-ORF2 region ("bs1", identified by miRanda, Supplementary Fig 4a). Thus, alternative approaches might be needed to obtain an unbiased view of functional let-7 binding sites in the L1 mRNA. Importantly, the binding site for the only miRNA targeting human LINE-1 mRNA described so far, miR-128, is also located in L1-ORF2[47]. We speculate that L1-ORF2, the largest and most conserved region among human LINE-1 subfamilies, is preferentially targeted by miRNAs because of the restricted space in the short L1 3′UTR, as it has been suggested by computational analysis for other mRNAs[76]. Accordingly, it was previously shown that knocking down the Microprocessor complex increased retrotransposition of a LINE-1 lacking the 3′UTR to the same extent as that of a full-length element[45]. Moreover, consistent with the mechanism suggested for gene silencing mediated by miRNA binding sites located in CDS[61,77], we have demonstrated that let-7 impairs translation of L1 ORF2p without affecting mRNA stability. This conclusion is further supported by the fact that no correlation between the levels of let-7 and the expression of L1Hs RNA was observed in human lung tumor samples (Supplementary Fig. 1b). Interestingly, the accumulation and translation of an L1 mRNA variant that has the natural "bs2rh" site substituted by a canonical let-7 8-mer site is similarly affected by let-7 overexpression as wild-type molecules (Supplementary Fig. 5j–m). This suggests that binding to the CDS region itself rather than the structure of base-pairing mediates translational repression, as previously described for other miRNA targeting CDS sites[61].

Since L1-ORF2p is expressed at a very low level and is essential for L1 retrotransposition[4], a small reduction in the abundance of this protein could unbalance RNP formation reducing human L1 mobilization. Besides binding L1-ORF2 sequences, let-7 could also be regulating other mRNAs encoding proteins that positively impact human L1 retrotransposition[59].

Notably, even though the let-7 miRNA-L1 mRNA interaction likely occurs in any cell that simultaneously expresses both RNAs, we did not observe any correlation between increased somatic L1 insertions and reduced let-7 levels in human breast cancer samples (Supplementary Fig. 1e). We speculate that in some cell types, other regulatory layers that suppress L1 mobilization at transcriptional or post-transcriptional levels[73] may overshadow miRNA-mediated L1 inhibition. Consistently, L1 reactivation is less frequent in breast cancer than in other type of tumors such a NSCLC[23,29,36], suggesting that in breast cancer additional mechanisms of control could be restricting L1 retrotransposition despite a reduced expression of let-7.

Let-7 is one of the most highly conserved families of miRNAs in the animal kingdom and is involved in multiple biological processes including differentiation, cell death, metabolism, and cancer[51]. Here, all our findings support a model in which let-7 also guides the RISC to the mRNA of active L1s and impairs L1-ORF2p translation, altering the ratio between L1-ORF2p and L1-ORF1p in the L1-RNP and consequently reducing LINE-1 retrotransposition (Fig. 6). Mature let-7 is highly expressed in differentiated cells[51], where different mechanisms repress L1 activity to avoid somatic L1 insertions[38,73]. We hypothesize that alterations in let-7 expression in human cancer lead to an increased mobilization of actively transcribed L1s and, moreover L1- mediated retrotransposition of non-autonomous transposable elements like Alu and SVA[35], increasing genome instability and contributing to tumor progression.

## Methods

**Sequencing data**. Both WGS (aligned to HG19), miRNA expression quantification and RNA-seq raw sequencing data files from TCGA were obtained from the Genomic Data Commons (GDC) Legacy Archive using the GDC Data Transfer Tool[78]. Cases of paired tumor-normal whole-genome sequencing (WGS) where tumor miRNAs expression data was available were retrieved for lung adenocarcinoma (LUAD) and lung squamous cell carcinoma (LUSC). High coverage (28-95x) WGS files aligned to hg19 from primary tumor and solid tissue normal samples, and miRNA gene quantification files from primary tumor were downloaded for LUAD (17 patients) and LUSC (28 patients).

**WGS analysis**. Putative somatic LINE-1 insertion calls for both normal tissue (NT) and primary tumor (PT) were obtained using MELT version 2.1.5[48]. To discard possible sequencing artifacts, candidate somatic insertions were further filtered including calls supported with a minimum of three split-reads, with the highest accuracy ASSESSMENT for breakpoint detection and passing all internal filters (MELT parameters ASSESS = 5 and FILTER = PASS). Polymorphic insertion calls were found using a curated database included in TEBreak software (https://github.com/adamewing/tebreak) and excluded from final results.

Several quality values were checked as a measure of filtering effects on original (unfiltered) MELT results. First, somatic insertions found in NT alone, and NT and PT simultaneously were expected to be zero, and only a maximum of one insertion was allowed for these values. All samples passed this additional filtering.

Furthermore, polymorphic L1 insertions after MELT filtering were controlled, requiring that a similar number was found for PT and NT samples, and that this number was uncorrelated with sample coverage. Four samples in LUSC (TCGA-60-2695, TCGA-60-2722) and LUAD (TCGA-55-1594, TCGA-55-1596) were excluded from analysis because only a low number (<10%) of polymorphic insertions passed all filters.

Filtered LINE-1 calls were considered tumor somatic insertions if detected in primary tumor filtered results and absent in unfiltered solid normal tissue insertion set within a range of 100 bp.

**Correlation with miRNA expression**. Samples were divided into two groups depending on whether putative somatic insertions were or were not found in the primary tumor. Only miRNAs with medium-high expression (over 100 reads per million (RPM) mapped reads) were considered. For some of the analysis, expression of specific miRNAs known to be involved in the development and progression of lung cancer was analyzed.

For each miRNA, outliers were discarded (we considered outliers values deviating more than two standard deviations from the mean in each group). Differentially expressed miRNAs were identified applying an unpaired two-tailed $t$ test adjusted by FDR = 1%. Results were confirmed using a rank-sum test. RPMs were normalized to the highest expression value of each miRNA to enable visualization of all miRNAs in the same graph. Tumor-suppressor miRNAs and oncomiRs related to lung cancer used for this analysis were described in a recent revision[50]. As a control, L1 insertion numbers were randomly reassigned to each sample and analysis was repeated. Moreover, analysis was done with L1 insertions determined by Helman and col. using Transpo-seq in lung and breast cancer samples obtained from TGCA as well[23]. Data processing and analysis were performed as described above.

**L1Hs RNA expression**. To analyze global TE expression in RNA-seq experiments, we use SQuIRE[55] (Software for Quantifying Interspersed Repeat Elements). SQuIRE quantifies expression at the subfamily level. It outputs read counts and fragments per kilobase transcript per million reads FPKM. Linear correlation between L1Hs RNA levels and miRNAs expression was calculated using Pearson Correlation Coefficient in GraphPad Prism 6.

**Cell culture**. HEK293T, PA-1, HeLa, and U2OS cells were originally obtained from ATCC and were provided by Drs Jose Luis Garcia-Perez (IGMM, Edinburgh, UK) and John V. Moran (University of Michigan, US). Lung cancer cell lines (A549, SK-MES-1) were provided by Dr Pedro Medina (GENYO, Spain). Stable Flp-In-293 cells expressing T7-tagged L1-ORF1p were previously generated for a different study[79].

HEK293T, HeLa, U2OS, A549, and SK-MES1 cells were cultured in high-glucose Dulbecco's Modified Eagle's Media (DMEM, Gibco) supplemented with GlutaMAX, 10% fetal bovine serum (FBS, Hyclone) and 100 U/mL penicilin–streptomycin (Invitrogen).

PA-1 cells were cultured in Minimal Essential Medium (MEM, Gibco) supplemented with GlutaMAX, 10% heat-inactivated FBS (Gibco), 100 U/mL penicilin-streptomycin (Invitrogen) and 0.1 mM Non-Essential Amino Acids (Gibco).

All cells were maintained in humidified incubators at 37 °C with 5% $CO_2$. The absence of *Mycoplasma spp.* in cultured cells was confirmed at least once a month by a PCR-based assay (Minerva). Cell identity was confirmed at least once a year using the STR genotype (at Lorgen, Granada).

**Retrotransposition assays**. Modified versions of previously established L1 retrotransposition assays[80–83] were performed and are described below[56]. In the Neo/Blast assays, $2 \times 10^5$ HeLa JVM cells were plated in 6-well tissue culture plates. Within 24 h, cells were co-transfected with 0.5-1 µg of L1 plasmid and 60 nM of let-7 mimic or 40 nM of let-7 inhibitor and their respective controls, scr, and c-, using Dharmafect DUO (Dharmacon) following manufacturer's instructions. For Neo assays, selection with 400 µg/mL of G418 (Life) was started 48 h post-transfection. For Blast assays, selection with 10 µg/mL of blasticidin (Millipore) was started 5 days post-transfection. In both cases, medium was changed every 2 days. Between 12 and 14 days after transfection, cells were washed with 1× PBS (Gibco), fixed (2% formaldehyde, 0.2% glutaraldehyde in 1× PBS), and stained with 0.5% crystal violet. Colonies were manually counted. The number of antibiotic-resistant colonies was used to quantify retrotransposition levels in cultured cells. Clonability assay was performed with 0.5 µg pU6i Neo and $1 \times 10^5$ HeLa cells[84].

Luciferase retrotransposition assays were performed as follows. $1 \times 10^5$ HEK293T/SK-MES-1 or $8 \times 10^4$ A549 cells were plated in 24-well tissue culture plates. Next day, 200 ng of pYX014, pYX015, or pYX017 were co-transfected with 60 nM of let-7 mimic or 40 nM of let-7 inhibitor (and their respective controls, scr, and c- respectively) using Lipofectamine 2000 (Life). Luciferase activity was measured 96 h post-transfection using Dual-Luciferase Reporter Assay System (Promega), in a GloMax Luminometer (Promega). Untransfected cells were used to correct for luciferase background.

EGFP-based retrotransposition assays were performed as follows. $4 \times 10^5$ HEK293T cells were plated in 6-well tissue culture plates. Next day, cells were co-transfected with 1 µg 99-UB-LRE3 and 40 nM of let-7 inhibitor or its control, c-, using Lipofectamine 2000 (Life). Retrotransposition (EGFP+ cells) was quantified 8 days post-transfection in a FACS Canto cytometer (BD).

**miRNAs mimics and inhibitors**. Let-7a/b mimic (C-300473-05 and C-300476-05), miR-34 mimic (C-300551-07), let-7a hairpin inhibitor (IH-300474-07), and their respective controls scr and c- (CN-002000-01-05 and IN-002005-01-05), were purchased from Dharmacon. They were resuspended in 1× siRNA Buffer (Thermo) to a working concentration of 20 µM and kept at −80 °C.

**RNA immunoprecipitation (RIP)**. $2 \times 10^6$ PA-1 cells were transfected in 10 cm tissue culture plates with 4 µg of FLAG-AGO2 and 25 nM of scr/mimic let-7 using lipofectamine 2000 (Life). Transfection with pBSKS (an empty plasmid) was used as a negative control for the IP. 48 h post-transfection, cells were washed with ice-cold 1× PBS, scraped, and transferred to a 1.5 ml tube. After centrifugation at $200g$ for 2 min, cells were resuspended in 200 µl of cold resuspension buffer (20 mM Tris (pH = 7.5), 150 mM NaCl, 1 mM EDTA, 1 mM EGTA) containing 1U/µL RNAsin Plus (Promega) and lysed adding 800 µl of cold lysis buffer (1% Triton X-100, 20 mM Tris (pH = 7.5), 150 mM NaCl, 1 mM EDTA, 1 mM EGTA, 1 mM phenylmethyl-sulfonyl fluoride (PMSF, Sigma), 1X cOmplete EDTA-free Protease Inhibitor cocktail (Roche) and incubating for 10–30 min on ice. After centrifugation (10,000$g$ for 10 min at 4 °C), 10 µL of RQ1 Dnase (Promega) was added to the supernatant. Immunoprecipitation of FLAG-AGO2 was performed with Dyna-beads Protein G (Life) and anti-FLAG M2 mouse (Sigma, F3165) for 3 h at 4 °C with rotation. After five washes with lysis buffer, 10% of sample-beads were used for protein extraction and western blot by adding LDS sample buffer and DTT and heating the samples at 70 °C for 20 min. The 90% of sample beads were incubated with RQ1 DNAse for 30 min for later RNA extraction with Trizol LS (Ambion).

**siCHECK luciferase assays**. In 24-well plates, $1 \times 10^5$ HEK293T cells were seeded per well. Within 24 h after seeding, cells were co-transfected with 10 ng of each siCHECK plasmid and 50–80 nM scr/let-7 mimic using Lipofectamine 2000 (Life). 24 h post-transfection, Firefly and Renilla luciferase measurements were performed in a GloMax Luminometer (Promega) using Dual-Luciferase® Reporter Assay System (Promega), following manufacturer's instructions.

**Site-directed mutagenesis**. Binding site mutant "bs2rhmut" was generated using an established protocol. Briefly, two sequential PCRs were performed, using an active L1.3 as a template. First, two PCRs were performed using the following primers under standard conditions: Let7-ORF2PCRafw/Let7-ORF2PCRa_PG2rv and Let7-ORF2PCRarv/Let7-ORF2PCRa_PG2fw. The products of both reactions were purified, mixed in equal amounts, and used as a template for a second PCR using primers Let7-ORF2PCRafw/ Let7-ORF2PCRarv. Conditions for this PCR were: 95 °C 5 min, 10 cycles with (95 °C 15 s, 50 °C 30 s, 72 °C 60 s), 25 cycles with (95 °C 15 s, 55 °C 30 s, 72 °C 60 s), 72 °C 10 min. The resulting product contained the mutated sequence in ORF2. This product was purified and cloned into a plasmid containing an active L1 (pJCC5/L1.3) using EcoNI and BsaBI sites, generating pJCC5/bs2mutL1.3. This mutant L1 was then cloned into pJM101 using NotI and BstZ17I sites, generating pJM101/bs2mutL1.3.

Binding site mutant 8-mer (pSA500 ORF2-8mer) was generated using the same protocol described above. For the first two PCRs, primers used were Let7-Bcl1-ORF2bs-PCRaFw/Let7-ORF2PCRa_8mer and Let7-ORF2PCRb_8mer/pCEP4_Rv. The products of both reactions were purified, mixed in equal amounts, and used as a template for a second PCR using primers Let7-Bcl1-ORF2bs-PCRaFw/

pCEP4_Rv. The product was purified and cloned into pSA500 using BclI and BstZ17I sites.

Restriction enzymes were purchased from New England Biolabs (NEB).

**Generation of 3′UTR variants of pSA500**. Sequences were ordered as oligos flanked by a BstZ17I site (See Supplementary Table 10). First, 1 µL of each Fw and Rv were annealed and phosphorylated with T4 Polynucleotide Kinase (PNK, NEB) using the following program: 30 min at 37 °C, 5 min at 95 °C, and then down to 25 °C at −5 °C/min. They were cloned in 3′UTR of pSA500 using a BstZ17I site, generating pSA500-3′UTR-scrb/bs2rh/8mer. Constructs were checked by digestion and Sanger sequencing.

**Western blot**. Cells were washed with 1× PBS, trypsinised and pelleted at 200$g$ for 4 min. To extract proteins, cell pellets were resuspended in 50–100 µL of RIPA buffer (Sigma) supplemented with 1× Complete EDTA-free Protease Inhibitor cocktail (Roche), PMSF (Sigma), 0.25% β-mercaptoethanol (Sigma) and incubated for 10 min on ice. Extracts were then centrifuged (18,000$g$ at 4 °C for 10 min) and debris-free supernatants were transferred to new tubes. Protein concentration was determined using the Micro BCA Protein Assay Kit (Thermo) following the manufacturer's instructions.

To control the transfection efficiency in pSA500 western blots, three fractions of the cells were pelleted in different tubes after trypsinization, and DNA, RNA, and protein extractions were performed simultaneously.

Proteins were resolved on an SDS-PAGE gel and transferred to a nitrocellulose membrane (BioRad). In L1-ORF2p western blots, proteins were resolved in a 4–15% Mini-PROTEAN TGX Precast Gels (BioRad), and transferred to a PVDF membrane using Trans-Blot Turbo Mini PVDF Transfer Packs (BioRad) and the Trans-Blot Turbo Transfer System (BioRad).

For western in Supplementary Fig. 5m, HEK-293T cells were transfected with plasmid constructs indicated and then selected for 1 week by hygromycin (100 µg/mL). Transfected cells were lysed and L1 RNP were enriched on sucrose cushion by ultracentrifugation. 3 µg of L1 RNP prep were loaded per lane and resolved in 4–10% gradient SDS-PAGE gels.

For blotting we used the following antibodies: a polyclonal rabbit anti-L1-ORF1p (1:1000, provided by Dr. Oliver Weichenrieder, Max-Planck, Germany), a polyclonal rabbit anti-L1ORF1p (1:1000, SE-6798), anti HMGA2 (1:1000, Abcam), anti DICER (1:1000, Cell Signalling), anti-tubulin (1:1000, Santa Cruz), anti-actin (1:10000, Sigma, A2228), anti-GFP (1: 2000, 3H9 clone, Chromotek). For chemiluminescent detection, we used anti-rabbit HRP (1:1000, Cell Signaling) or anti-mouse HRP (1:1000, Cell Signaling), and Clarity ECL Western Blotting Substrate (BioRad) or SuperSignal West Femto Maximum Sensitivity Substrate (Thermo). Images were acquired with an ImageQuant LAS4000 and quantified using ImageJ software. For Odyssey analysis, anti-rabbit and anti-mouse fluorescent antibodies (LI-COR) were used at 1:10000 dilution, and detection and quantification were performed in Odyssey (LI-COR).

**qPCR and RT-qPCR**. To control plasmid transfection in L1-ORF2p western blots, transfected cells were lysed in a buffer containing 10 mM Tris pH=8.2, 10 mM EDTA, 200 mM NaCl, 0.5%SDS and 200 µg/ml proteinase K, and incubating for 3 h at 56 °C with agitation. Afterwards, DNA was purified with phenol:chloroform: isoamyl alcohol (25:24:1, Thermo) following standard procedures. 50 ng of each DNA sample were used per qPCR reaction (GoTaq qPCR Mix, Promega), and an untransfected control was used to discard plasmid contamination. qPCR method is as follows: 5 min at 95°, and 40 cycles of 15 s at 95° followed by 60 s at 60 °C. Plasmid levels were quantified using CMV and EBNA-1 primers, and normalization was performed using genomic GAPDH primers.

RNA was extracted from cells using Trizol (Invitrogen), following the manufacturer's instructions. 1 µg of RNA was subsequently treated with RQ1 DNAse and purified by a phenol/chloroform extraction. cDNA was synthesized using High-Capacity cDNA Reverse Transcription Kit (Applied Biosystems), and used for qPCR (GoTaq qPCR Mix, Promega) using standard protocols. Two controls were used to verify the absence of contaminating gDNA: no-RT and no-template. Endogenous L1 mRNA was quantified using N51 primers[85]. Transfected L1 mRNA was quantified using NEOjunct2 primers designed to exclusively amplify the spliced NEO cDNA[86] (in Supplementary Fig. 5a, b) or SV40 primers when the NEO cassette was absent (Fig. 5c and Supplementary Fig. 5h, k). GAPDH was used to normalize in Fig. 5c because an additional qPCR was used to quantify plasmid levels and discard differences in transfection efficiency. Elsewhere, EBNA mRNA, expressed from the backbone of the plasmids, was used to normalize.

For mature miRNAs quantification, a RT-qPCR was used. 1 µg of total RNA isolated with Trizol was polyadenylated and then cDNA was synthesized, using qScript miRNA cDNA synthesis Kit (QuantaBio). Quantitative PCR was performed using a universal primer against poly(A) and a miRNA-specific primer that allows the specific detection of polyadenylated mature miRNA and not its precursors (QuantaBio). qPCR was performed with primers Let-7a, Let-7b, and miR-34a.

**Flow cytometry**. In 6-well plates, $2 \times 10^5$ HeLa cells were seeded per well. Next day, cells were transfected with 1 µg of pVan583 and 40 nM c-/let7inh using

lipofectamine 2000 (Life). Seventy-two hours post-transfection, cells were washed with 1× PBS (Life), detached with TrypLE Express (Gibco) for 5–10 min at 37 °C, pelleted 4 min at 200g, resuspended in 1× PBS with 5% FBS and 5 mM EDTA, and passed through a 70 μM filter. After incubation with 10ug/mL 7AAD (Sigma) for 10 min, fluorescence was quantified in a FACSAria (BD) cytometer. For each replicate, $10^5$ cells were passed through the cytomerer. Only live and transfected cells (7AAD⁻ and GFP⁺, between 3600 and 9300 cells) were used for %mCherry analysis, which was performed using FlowJo software (LLC) version 10. Controls were used to set the threshold for each fluorescent channel of detection: untransfected cells, and cells expressing either GFP only or mCherry only.

**Confocal microscopy**. $2 \times 10^5$ U2OS per well were seeded in 6-well tissue culture plates. Next day, cells were transfected with 1 μg of DNA and 60 nM scr/mimic using lipofectamine 2000 (Life), following standard protocols. Twenty-four hours post-transfection, cells were detached and re-seeded in 24-well plates where a UV-sterilized glass slide had previously been placed. Forty-eight hours post-transfection, cells were washed in 1× PBS, fixed in paraformaldehyde (4% in 1× PBS) for 30 minutes at room temperature, and slides were mounted with Slow-Fade Gold Antifade reagent with DAPI (Life). Slides were imaged using a Zeiss LSM-710 confocal microscope (Leica).

**Hybridization between let-7 and L1 prediction**. The potential structure formed by let-7 and WT L1 or its binding site mutant was analyzed by RNAHybrid[69]. Briefly, the region of L1Hs identified as "bs2" with RNA22 was paired to the mature sequence of let-7, using default parameters.

**Plasmids**. pYX014, pYX015 and pYX017[83], JM101/L1.3[70], JM105/L1.3[4], JJ101/L1.3[87] and TAM102/L1.3[80], 99-UB-LRE[88], Tgf21-Neo[89], Zfl2-1-Neo, and Zfl2-2-Neo[90] have been previously described. JJ101/L1.3Δ3'UTR is a derivative of JJ101/L1.3 but the L1 lacks the 3'UTR[45]. JM101/L1.3bs2mut was generated by cloning a bs2mut L1 into JM101/L1.3 (cloning strategy described in the "Site-directed mutagenesis" section of the methods). FLAG-Ago1 and FLAG-Ago2 were a gift from Edward Chan (Addgene plasmid #21533 and #21538)[91]. pSA500 is as pAD500[63] where the TAP epitope was replaced by 3 consecutive copies of the FLAG epitope tag obtained from PJCC5 ORF1T7 ORF3XFLAG using BclI and BstZ17I restriction enzyme sites. pVan583 is a derivative of JM101/L1.3 where EGFP and mCherry were cloned in frame with the last amino acid of L1-ORF1 and L1-ORF2, respectively. All psiCHECK2 constructs were generated by cloning sequences synthesized and cloned in pUC57 (Genescript) into psiCHECK2 using XhoI and NotI restriction enzyme sites (Promega).

**Reporting summary**. Further information on research design is available in the Nature Research Reporting Summary linked to this article.

## Data availability

Data sets used in Fig. 1 and Supplementary Fig. 1 are detailed in Supplementary Table 1. All data are available from the GDC legacy archive (https://portal.gdc.cancer.gov/legacy-archive). Though most data files can be accessed without requiring access approval, WGS files need a special request due to their potential identification information. Researchers interested in accessing restricted data can obtain authorization following the instructions in https://gdc.cancer.gov/access-data/obtaining-access-controlled-data. Uncropped versions of the western blots are provided in Supplementary Fig. 6. The data supporting the findings of this study are available from the corresponding authors upon reasonable request. Source data are provided with this paper.

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

## Acknowledgements

The results shown here are in part based upon data generated by the TCGA Research Network: https://cancer.gov/TCGA. We thank Jose Luis Garcia-Perez (IGMM, Edinburgh, UK) for his valuable contribution to this project. We are also grateful to Marie-Jeanne Kempen, Jose Luis Garcia-Perez and Javier F. Caceres (IGMM, Edinburgh, UK) for comments and critical reading of the manuscript. We thank current members of the lab for useful discussions and methodological support. We are grateful to Drs. Jose Luis Garcia-Perez and John V. Moran (University of Michigan, US) for sharing numerous human L1 vectors and cell lines; Aleksandra Helwak for valuable input at the beginning of this project, Wenfeng An (Dakota State University, USA) for sharing L1-Fluc plasmids; Aurelien Doucet for sharing pAD500; Norihiro Okada and Masaki Kajikawa (Tokyo Institute of Technology, Japan) for sharing tagged zebrafish LINE-2 plasmids; Haig Kazazian and John Goodier (John Hopkins, US) for sharing tagged mouse LINE-1 plasmids; Tomo-ichiro Miyoshi (Kyoto University) for sharing pJCC5ORF1T7-ORF3xFLAG plasmid; Pedro Medina (Genyo, Spain) for the lung cancer cell lines and technical advises; Oliver Weichenrieder (MPI Tubingen, Germany) for the polyclonal L1-ORF1p antibody; Ian Adams (IGMM, Edinburgh, UK) for Flp-In-293 cells expressing T7-tagged ORF1p; Verónica Ayllón (Genyo, Spain) for support with flow cytometry analysis; and Guillermo Barturen (Genyo, Spain) for support with bioinformatic and statistical analysis. We also thank Jennifer Parra for technical support, and Genyo's flow cytometry unit. The work in the lab of G.C. was supported by the European Research Council (ERC-2009-StG 243312, Retrogenomics), by the French Government (National Research Agency, ANR) through the "Investments for the Future" programs LABEX SIGNALIFE ANR-11-LABX-0028-01 and IDEX UCAJedi ANR-15-IDEX-01, by CNRS

(GDR 3546), and by the University Hospital Federation (FHU) OncoAge.This article is part of the doctoral thesis of P.T.-R., a graduate student in the PhD programme of Biomedicine at the University of Granada, who was supported by MINECO (PEJ-2014-A-31985 and SAF2015-71589-P). S.R.H. is funded by MINECO cofounding by European Regional Development Fund (SAF2015-71589-P), Ramón y Cajal (RYC-2016-21395) and a Career Integration Grant-Marie Curie (FP7-PEOPLE-2011-CIG-303812).

## Author contributions

S.R.H conceived and supervised the study. P.T.-R. and S.R.H designed and interpreted the experiments. P.T.-R. performed most of the experiments and data analysis. A.R.-R and G.P. provided all the bioinformatic analysis, including identification of somatic L1 insertions by MELT. P.T.-R and G.P. performed statistical analysis. A.R.-R was responsible for data management. L.S. provided technical support and performed experiments. S.A.-C. generated pSA500 construct. S.V. and G. C. generated and characterized the pVan583 construct. P.T.-R., G.P., and S.R.H wrote the manuscript and all the authors contributed with a critical reading.

## Competing interests

The authors declare no competing interests.
