## [Peer Review File · Nature Communications]

Reviewers' comments:

Reviewer #1 (Remarks to the Author):

This manuscript describes a regulatory role for the miRNA let-7 in the translation of L1 retrotransposons - and specifically, the ORF2 protein. The regulation of L1 is a multi-tiered process with complex interactions occurring at multiple lifecycle stages. Elucidating L1 regulators is a challenging and growing area of contemporary molecular biology research. Few details regarding mechanisms of ORF2p translational control exist; thus, this paper brings additional information to bear. Although skepticism regarding the specific role(s) of L1-host interactions in cancer development is warranted, this manuscript points to a bona fide effect: steady state gene expression changes in cancer - such as let-7 miRNA levels - may contribute to steady state changes in ORF2p and an increase in retrotransposition rate.

The manuscript is generally well written and attention is paid to the interpretation of results and logical narrative flow - however, additional copy editing for grammatical English will still be useful in a few places (I have not enumerated them - e.g. in several places the word revised is used in place of reviewed). Also, I found evaluating the supplemental data to be frustrating (elaborated below).

I provide the following suggested revisions prior to publication:

PAGE 2: "...giving rise to hundreds of molecules of L1-ORF1p molecules [sic] and apparently one molecule of L1 ORF2p that binds back..."

 to my knowledge the stoichiometry of ORF1p to ORF2p in L1 RNPs remains to be definitively shown. While I believe that hundreds of ORF1s may be translated, it is not clear that they join a singular, discrete RNP.

PAGE 2: "...may hijack the L1-encoded proteins..."

 while Alus may be considered a parasite's parasite, 'hijacking' the ORF2 protein, mRNAs are probably not streamlined for co-opting ORF2p and most processed pseudogenes are presumed accidental byproducts. Language may benefit from more nuancing here.

(Major) Supplementary materials: part of the supplement is in rather poor shape - what is referred to as SUPP FIG. 6 is actually a collection of different blots/images. It's not at all acceptable to just data-dump a bunch of blots. If the authors want to include these data - and they should do so - then they should also be properly formatted and labeled so that they are readily readable and the significance / identity of each lane is apparent. I would not allow these data to be published in the current form.

Although "mock" IPs are referred to in multiple places, used as controls for IPs - the definition of this mock is never given, neither in figure legends nor in methods. In most cases this is probably the same IP in the absence of transfection, or similar - but in each case the control must be clearly defined.

PAGE 16: "Strikingly, bs2mut L1.3 retrotransposition was less effected..."

 Isn't this the expected result? Why is this striking?

PAGE 18/19: Experiments related to fluorescent-tagged ORF proteins. The authors acknowledge that fluorescent tagging compromises L1 retrotransposition - and this has been reported by others previously for GFP-tagged ORF proteins. However, this data can (and should) be instead presented in the supplement, given the fact that interpretation may be significantly complicated by the decreased activity (it can be substantially decreased). I believe the whole experiment (data currently not shown and related figures) should be moved to the supplement. Whether it is ultimately retained as a main figure or moved to the supplement - Fig 5e is too small to comfortable view and should be enlarged.

Figure 6: micro-RNA schematic above let-7 is too small and, does it make sense to represent RISC as bound to the mRNAs in the absence of let-7 - it looks as through RISC binds independent of let-7. Also, even though a description of the intentions of this figure are given in the main text, this figure should have a concise descriptive caption as a legend. The "oncogenic protein" part of the schematic could simply be omitted - since the paper is dealing with the effects of let-7 on L1 and let-7 effects on other mRNAs are described elsewhere.

Several different cell lines are used throughout this study - in all figure legends it should be made clear the cell type used. In Figure 3 I think both HeLa and PA-1 were (used 3d - described on page 14), but in the figure this was not clear - other similar instances should be made unambiguous.

Signed: John LaCava

Reviewer #2 (Remarks to the Author):

Authors presented a handful number of miRNAs that control the activity of human-specific LINE1 elements in translational level. The miRNAs are let-7a, e, and miR-34a that are downregulated in cancers

where at least one L1 is somatically inserted to the genome. By reanalyzing whole genome sequencing data of 41 NSCLC patients samples along with tumor miRNA-seq data from TCGA, they found hundreds of tumor-specific L1 retrotransposon insertions using the MELT. The level of let7a was strongly associated with the insertions and negatively regulates human L1-retrotransposition in vitro. They found let-7 binds directly the coding region of L1 mRNAs rather UTRs and the binding sites seem to be noncanonical lacking seed-sites. The swapping the sites to non-related sequence or mutagenesis of it showed that let-7 is controlling L1 expression in translation through the noncanonical sites.

1) By now, the miRNA is known to repress target genes in both translational and post-transcriptional levels through a well-characterized seed-type sites (Agarwal, 2015, eLife) and a few noncanonical target sites were reported by systematic analyses (Kim et al., 2017, Nature Genetics). We also know a few noncanonical miRNA target sites were reported to be functional. Nevertheless, the let-7 sites in L1ORF2 includes a wobble pair and two-nucleotide bulge in the seed site, which are not reported to be functional previously, although the folding energy b/w let-7 and target site is strong. In addition, it is still unclear whether this noncanonical site is functional (translational repression) due to the location (ORF) or regardless of the location or shape of the base-pairing or others. If they move the site to the 3'UTR, author think is also functional?

2) Related to the 1), what is the specific mechanism of the translation regulation? is this regulation also working in the other protein-coding genes? Are they require the bulge sites or GU wobble site to repress translation only? Although they used a perfect match site and mutagenesis on the site, they need to mutagenesis base-by-base (or basepair) to clearly understand why they observed only translational repression. If they can show the mechanism, it would be very interesting.

3) Related to 2), authors also showed the L1 RNA was not downregulated by let-7a, nevertheless. I wonder whether they found the downregulation of L1 RNAs via the perfect bs as they found that let-7 mimic showed more repression (Fig 4b).

3-1) After overexpression of let-7a or down-regulation of let-7a, did they also detect the changes of canonical targets (with 7-mer sites) of let-7 ?

3-2) I wonder whether the 5'isomir of let-7a can include a potential target sites in L1.

4) Unclear how they normalized the miRNA level in Fig 1b. I wonder a rational why they used the maximum value in each sample to normalize? Each sample could have different maximum level of miRNA actually and forcing to normalize the level of miRNA using the maximum value could include unexpected artifact as well. How the results are different if they use the rank-sum test on the unnormalized level of miRNAs?

5) As previously reported, newly inserted retrotransposons are often inactivated by epigenetic way. Please the check the L1 RNA level (using RNA-seq) and methylation level in L1 to find the association with the L1 insertion or miRNA levels.

6) Fig 3d, L1 and HMGA2 enrichment experiments have to be repeated and display the standard deviations of the changes.

7) There are more cancers that are involved with genomic instability. I wonder if this let-7 regulation for L1 is also true in other cancer.

Reviewer #3 (Remarks to the Author):

In the manuscript titled "The tumor suppressor microRNA let-7 inhibits human LINE-1 retrotransposition" Tristán-Ramos et. al. explored the role of microRNA let-7 in L1 retrotransposition. With well-designed experiments, this study demonstrates that let-7 regulates L1 transposition by inhibiting ORF2p translation and such inhibition is achieved via a potential let-7 binding site within ORF2p, thus providing novel insights into the role of let-7 in maintaining genome integrity. The manuscript is informative and well written, specific comments are listed below:

Figure 1 and supplementary figure 1 illustrated the changes in miRNA expression in tumor samples with or without L1 insertions. What are the miRNA expression levels of the matching normal samples? Do patients with L1 insertion also have low let-7 expression in normal samples or such phenomenon can only be observed at the tumor samples?

The use of both let-7 inhibitors and mimic reasonably demonstrates that let-7 inhibits L1 retrotransposition. However, it would be nice to demonstrate or comment on the specificity of let-7 inhibitors (both within and across let-7 families).

Page 10 "We were able to recapitulate this effect in a different cell line, HEK293T..", this indicates both let-7 and miR-34a results were recapitulated. What is the endogenous expression level of miR-34a in HEK293T? What's the effect of over-expressing miR-34a in HEK293T? Results need to be included at least as supplementary figures.

Page 12 "we performed the luciferase-based retrotransposition assay in two lung cancer cell lines with significantly different endogenous levels of let-7... depletion of let-7 increased L1 retrotransposition by 2.5 times on average (Fig. 2d)" Do the two cell lines exhibit different L1 retrotransposition activity under control condition? And in general, does endogenous let-7 expression correlate with L1 retrotransposition activity?

Figure 5e, quantification needed (something similar as performed in 5d)

POINT BY POINT RESPONSE TO REVIEWERS

Reviewer #1

This manuscript describes a regulatory role for the miRNA let-7 in the translation of L1 retrotransposons - and specifically, the ORF2 protein. The regulation of L1 is a multi-tiered process with complex interactions occurring at multiple lifecycle stages. Elucidating L1 regulators is a challenging and growing area of contemporary molecular biology research. Few details regarding mechanisms of ORF2p translational control exist; thus, this paper brings additional information to bear. Although skepticism regarding the specific role(s) of L1-host interactions in cancer development is warranted, this manuscript points to a bona fide effect: steady state gene expression changes in cancer - such as let-7 miRNA levels - may contribute to steady state changes in ORF2p and an increase in retrotransposition rate.

The manuscript is generally well written and attention is paid to the interpretation of results and logical narrative flow - however, additional copy editing for grammatical English will still be useful in a few places (I have not enumerated them - e.g. in several places the word revised is used in place of reviewed). Also, I found evaluating the supplemental data to be frustrating (elaborated below).

We apologize for the presence of several grammar errors; a thorough revision of grammar English has now been done in the revised manuscript. Also, the supplemental data have been edited and organized to make it more comprehensive (details below).

I provide the following suggested revisions prior to publication:

PAGE 2: "...giving rise to hundreds of molecules of L1-ORF1p molecules [sic] and apparently one molecule of L1 ORF2p that binds back..."

 to my knowledge the stoichiometry of ORF1p to ORF2p in L1 RNPs remains to be definitively shown. While I believe that hundreds of ORF1s may be translated, it is not clear that they join a singular, discrete RNP.

We acknowledge and agree with the point raised by the reviewer. We have modified this accordingly in page 2. The sentence now stands: '...giving rise to L1-ORF1p and L1-ORF2p that bind preferentially to the same L1 mRNA to form a ribonucleoparticle (RNP) (Kulpa et al. 2006)'. We have also added a sentence in the same page "However, ORF2p is expressed at a significantly lower level than ORF1p (Ardeljan et al. 2020; Taylor, et al. 2013), and these differences are thought to be controlled at the level of translation (Alisch et al. 2006)"

PAGE 2: "...may hijack the L1-encoded proteins..."

 while Alus may be considered a parasite's parasite, 'hijacking' the ORF2 protein, mRNAs are probably not streamlined for co-opting ORF2p and most processed pseudogenes are presumed accidental byproducts. Language may benefit from more nuancing here.

We apologize for the lack of precision in the terminology. We have modified this sentence accordingly in page 2: "Other non-autonomous retrotransposons such as Alu and SINE-R/VNTR/Alu (SVA) may hijack the L1-encoded proteins and be mobilized in trans (Dewannieux et al. 2003, Hancks et al. 2011). Furthermore, L1-encoded proteins can sporadically generate pseudogenes using cellular mRNAs as templates (Esnault et al. 2000)".

(Major) Supplementary materials: part of the supplement is in rather poor shape - what is referred to as SUPP FIG. 6 is actually a collection of different blots/images. It's not at all acceptable to just data-dump a bunch of blots. If the authors want to include these data - and they should do so - then they should also be properly formatted and labeled so that they are readily readable and the significance / identity of each lane is apparent. I would not allow these data to be published in the current form.

We apologize for the lack of clarity. We have now labelled all the blots appropriately so that every lane and every band can be easily identified in Supp Fig. 6. Furthermore, to improve supplementary materials, a schematic representation of the bioinformatic analysis used to generate each graph has been included in Supp. Fig. 1.

Although "mock" IPs are referred to in multiple places, used as controls for IPs - the definition of this mock is never given, neither in figure legends nor in methods. In most cases this is probably the same IP in the absence of transfection, or similar - but in each case the control must be clearly defined.

We have clarified this issue in the figure legends and the methods section. For all IPs, 'mock' condition refers to cells transfected with an empty plasmid.

PAGE 16: "Strikingly, bs2mut L1.3 retrotransposition was less effected..."

 Isn't this the expected result? Why is this striking?

We agree with the reviewer that a reduction of let-7 effect on bs2rhmut L1.3 retrotransposition was the expected result, therefore we have removed "striking" in that sentence.

PAGE 18/19: Experiments related to fluorescent-tagged ORF proteins. The authors acknowledge that fluorescent tagging compromises L1 retrotransposition - and this has been reported by others previously for GFP-tagged ORF proteins. However, this data can (and should) be instead presented in the supplement, given the fact that interpretation may be significantly complicated by the decreased activity (it can be substantially decreased). I believe the whole experiment (data currently not shown and related figures) should be moved to the supplement. Whether it is ultimately retained as a main figure or moved to the supplement - Fig 5e is too small to comfortable view and should be enlarged.

We have now followed these suggestions and included data previously not shown in Supp Fig 5 and clarified in the revised text (page 22): "First, we confirmed that ORF1p-

GFP was expressed from this construct (**Supplementary Fig. 5n**) and that this tagged L1 was able to retrotranspose, although the addition of both fluorescent tags reduced its activity to ~30% of its untagged counterpart, JM101/L1.3 (**Supplementary Fig. 5o**)”.

We believe this reduction in the retrotransposition activity does not compromise the interpretation of our data, since we used this construct to detect changes in the percentage of ORF1p-GFP positive cells expressing ORF2p-Cherry protein. To clarify this point we have included a representative FACS histogram for each condition in Fig 5d. Regarding the confocal microscopy experiment, it has been moved to **Supplementary Fig. 5q** and enlarged.

Figure 6: micro-RNA schematic above let-7 is too small and, does it make sense to represent RISC as bound to the mRNAs in the absence of let-7 - it looks as though RISC binds independent of let-7. Also, even though a description of the intentions of this figure are given in the main text, this figure should have a concise descriptive caption as a legend. The "oncogenic protein" part of the schematic could simply be omitted - since the paper is dealing with the effects of let-7 on L1 and let-7 effects on other mRNAs are described elsewhere.

We apologize for making the image of let-7 microRNA too small, but we had not represented RISC without let-7. We have now coloured the miRNA in red to remark that it is loaded in RISC and increased its size. Besides, we have added a detailed description in the legend of the figure to clarify it.

Regarding the last point, the oncogenic protein part of the image is shown to illustrate that we uncover a novel tumor suppressor role for let-7 apart from regulating oncogenes, something that maybe some readers may not be aware of. Thus, we believe that it is useful. However, to remark that this effect of let-7 is not described in the paper we have now drawn it in grey and included a sentence in the legend.

Several different cell lines are used throughout this study - in all figure legends it should be made clear the cell type used. In Figure 3 I think both HeLa and PA-1 were (used 3d - described on page 14), but in the figure this was not clear - other similar instances should be made unambiguous.

We have carefully revised and clarified which cell line has been used for every experiment, both in the text and in the figure legends (e.g. legend of Figure 5).

Reviewer #2:

Authors presented a handful number of miRNAs that control the activity of human-specific LINE1 elements in translational level. The miRNAs are let-7a, e, and miR-34a that are downregulated in cancers where at least one L1 is somatically inserted to the genome. By reanalyzing whole genome sequencing data of 41 NSCLC patients samples along with tumor miRNA-seq data from TCGA, they found hundreds of tumor-specific L1 retrotransposon insertions using the MELT. The level of let7a was strongly associated with the insertions and negatively regulates human L1-retrotransposition in vitro. They found let-7 binds directly the coding region of L1 mRNAs rather UTRs and the binding sites seem to be noncanonical lacking seed-sites. The swapping the sites to non-related sequence or mutagenesis of it showed that let-7 is controlling L1 expression in translation through the noncanonical sites.

- 1) *By now, the miRNA is known to repress target genes in both translational and post-transcriptional levels through a well-characterized seed-type sites (Agarwal, 2015, eLife) and a few noncanonical target sites were reported by systematic analyses (Kim et al., 2017, Nature Genetics). We also know a few noncanonical miRNA target sites were reported to be functional. Nevertheless, the let-7 sites in L1ORF2 includes a wobble pair and two-nucleotide bulge in the seed site, which are not reported to be functional previously, although the folding energy b/w let-7 and target site is strong. In addition, it is still unclear whether this noncanonical site is functional (translational repression) due to the location (ORF) or regardless of the location or shape of the base-pairing or others. If they move the site to the 3'UTR, author think is also functional?*

We acknowledge the point raised by reviewer. The luciferase experiment and retrotransposition assay carried out with mutated L1bs2 (Fig. 4) showed that the nucleotides substituted in this sequence disrupted the interaction of let-7 with a functional binding site in ORF2. However, we agree with the reviewer that the interaction between let-7 and bs2 predicted by RNA22 program was not expected to generate a functional binding site. We have carefully reanalysed the L1mRNA region containing bs2 with RNAhybrid software. **Interestingly, this software found an offset 7-mer binding site type, described to be functional by Kim et al. 2016, overlapping almost entirely with the bs2 predicted by RNA22 program.** Although both of them are theoretical binding sites, the interactions described by RNAHybrid suggest that this sequence is a *bona fide* let-7 binding site. The updated pairing between L1Hs and let-7 is shown in Fig 4a as predicted by RNAHybrid and has been called 'bs2rh'. Moreover, the pairing between bs2rhmut and let-7 is shown in Supp Fig 4b. The text has been modified accordingly in page 16 and 17:

“A deeper analysis of the residues in this region interacting with let-7 microRNA using RNAhybrids software (Rehmsmeier et al. 2004) suggests that the functional 'bs2' is located within the coding sequence of L1-ORF2 (position 4587-4610 in L1.3, a commonly used human RC-L1, accession code # L19088.1 (Sassaman (1997)), between the RT and Cysteine-rich domains of this protein (Fig. 4a, left panel). **Importantly, it is predicted to form a duplex with let-7 miRNA consisting of seven Watson–Crick pairings at positions 3-9 followed by an adenine at the mRNA nucleotide corresponding to the first nucleotide position of the miRNA, resembling a previously described functional noncanonical binding site termed offset 7-mer (Kim et al. (2016)) (Fig. 4a).** Altogether, these results suggest that this refined binding site, hereafter referred to as 'bs2rh' is a *bona fide* let-7 binding site. To further validate this binding site ('bs2rh') we generated a mutant sequence ('bs2rhmut', see Fig. 4b, left panel).”

This result has also been comment in the discussion section: “In fact, we have demonstrated that ORF2 contains a noncanonical offset 7-mer let-7 binding site previously described as functional site for microRNA targeting.”

Regarding the last question: **in our psiCHECK2 experiments, where the binding site is located five times in tandem in the 3'UTR of a luciferase gene (Fig. 4b), we do see a functional effect measured as a reduction of the luciferase activity.** Furthermore, this effect was abolished in a mutated version of the binding site, 'bs2rhmut'. Moreover, to analyse if the translational repression effect described in our work is dependent of the binding site location and its native context, **we have now cloned bs2rh in the 3'UTR of**

ORF2p using the pSA500 plasmid, as well as a mutated bs2rh that generates a canonical 8-mer site (Supp Fig. 5g). It is worth mentioning that for this experiment the binding site was cloned once for comparison purposes, whereas in the aforementioned luciferase experiment (Fig.4b), the binding site was cloned five times. We have now included these results in page 20 and as new panels in Supplementary Fig.5:

“To understand whether let-7 mediated translational repression of ORF2p is due to the specific interactions with the offset 7-mer site or to its location within the coding sequence, we generated three variants of pSA500 in which we introduced different sequences in its 3’UTR : a scrambled sequence (‘scrb’), the binding site (‘bs2rh’) and a modified bs2 that contains a canonical 8-mer site for let-7 (‘8mer’) (**Supplementary Fig. 5g**). We co-transfected all these constructs in HeLa cells with let-7 mimic. First, by RT-qPCR we observed that similar levels of transfection (measuring constitutive EBNA expression from the plasmid backbone, **Supplementary Fig. 5h**) and let-7 overexpression (measuring the effect on endogenous DICER, **Supplementary Fig. 5h**) were achieved. **The levels of ORF2 mRNA were not significantly affected in any case, although we observed a tendency towards a reduction on the RNA levels upon placement of the binding site (‘bs2rh’) or the modified 8-mer binding site (‘8mer’) in the 3’UTR (Supplementary Fig. 5h).** Furthermore, western blot analysis showed that placement of ‘bs2rh’ sequence in the 3’UTR of pSA500 slightly enhanced the reduction of ORF2-F protein upon let-7 overexpression (**Supplementary Fig. 5i**), an effect that was more prominent when the canonical site (‘8mer’) was tested. In agreement with previous studies (Kim et al. 2016), these results suggests that the proficiency of ‘bs2rh’, a noncanonical offset 7mer site, is weaker than that of a canonical let-7 binding site when they are located in 3`UTR. Moreover, we cannot rule out that the translational repression mediated by both biding sites located in 3`UTR could be attributed to mRNA destabilization”.

2) Related to the 1), what is the specific mechanism of the translation regulation? is this regulation also working in the other protein-coding genes? Are they require the bulge sites or GU wobble site to repress translation only? Although they used a perfect match site and mutagenesis on the site, they need to mutagenesis base-by-base (or basepair) to clearly understand why they observed only translational repression. If they can show the mechanism, it would be very interesting

Following the reviewer’s recommendation, and to understand if the specific translational repression effect mediated by this offset 7-mer site requires the structure of this non canonical binding site, **we have generated two point-mutations in the ORF2 coding region to transform this binding site into a canonical let-7 8-mer site.** Unfortunately, mutagenesis base-by-base of this binding site is clearly limited by the location within the coding sequence. We have now included these new results in page 21 and as new panels in Supplementary Fig.5:

“Additionally, using site directed mutagenesis we introduced two point-mutations in the ORF2 coding region to transform the offset 7-mer into a canonical let-7 8-mer site, generating pSA500-ORF2-8mer (**Supplementary Fig. 5j**). We co-transfected this construct in HeLa cells with let-7 mimic. Interestingly, the 8-mer site within the ORF does not affect the levels of mRNA (**Supplementary Fig. 5k**) and leads to a decrease in the protein level similar to that observed above for ‘bs2rh’ (**Supplementary Fig. 5m**). Altogether, these results suggest that the translational repression mediated by bs2rh

mainly depends on its location within the coding sequence, rather than on its non-canonical interaction.”.

Our results are in agreement with previous studies (Hausser et al. (2013) Genome Research) suggesting that sites located in the CDS are most potent in inhibiting translation than those located in the 3'UTR independently of sequence and structure properties. Consequently, it is probable that the CDS binding sites repress translation through a specific mechanism as it has been suggested by other authors (Zhang et al (2018) Nat Struct Mol Biol (25(11):19-1027). This is now included in page 26 of the discussion:

“Interestingly, the accumulation and translation of L1 an mRNA variant that have the natural 'bs2rh' site substituted by a canonical let-7 8-mer site is similarly affected by let-7 overexpression as wild-type molecules (Supplementary Fig. 5j-m), suggesting that binding to the CDS region itself rather than the structure of the of base-pairing mediates translational repression, as previously described for other miRNA targeting CDS sites (Hausser, et al. 2013)).”

3) Related to 2), authors also showed the L1 RNA was not downregulated by let-7a, nevertheless. I wonder whether they found the downregulation of L1 RNAs via the perfect bs as they found that let-7 mimic showed more repression (Fig 4b).

We demonstrated that **endogenous** L1RNA was not downregulated by let-7 (Fig.5a) neither ORF2L1 mRNA from a plasmid (pSA500) (Fig. 5c, bottom panel). To further corroborate that, we **have now included a qRT-PCR experiment showing that the levels of a full-length bicistronic L1 mRNA overexpressed from a plasmid (JM101/L1.3) do not change upon let-7 overexpression or depletion.** We have included in the text in page 18: “Similarly, L1 mRNA levels were not decreased upon let-7 overexpression (**Supplementary Fig. 5a**) or increased upon let-7 depletion (**Supplementary Fig. 5b**) when L1 was overexpressed in HEK293T cells.”

None of the software used found a canonical let-7 binding site in L1 sequence. It is worth mentioning that in Fig. 4b we cloned a let-7 perfect binding site (**i.e. 21 nt match**) in the 3'UTR of a luciferase gene as a positive control. Considering that the interaction between let-7 and a perfect binding site could function as an siRNA instead of as a miRNA, **to determine the effect of a canonical let-7 binding site on L1 mRNA an 8-mer site was cloned in the 3'UTR and on the ORF2 coding sequence using pSA500 plasmid ((Supp. Fig. 5g) and (Supp Fig 5j)).** As stated in point 2 and point 3, **8-mer site in CDS does not mediate ORF2p RNA downregulation.** However, when this site was located in the 3'UTR, although not significant, there was a decrease in the levels of ORF2 mRNA. Of note, we would not expect massive differences as it is only one site whereas classic targets of let-7 contain several sites in their 3'UTR (i.e. HMGA2 contains seven let-7 binding sites). **We conclude that a let-7 canonical site in ORF2 doesn't lead to mRNA degradation while cannot exclude this possibility when it is located in 3` UTR.** These results have been commented in the revised manuscript (see answers to 2) and 3)).

3-1) After overexpression of let-7a or down-regulation of let-7a, did they also detect the changes of canonical targets (with 7-mer sites) of let-7?

We apologize for the lack of clarity. Indeed, **we found changes in two canonical let-7 targets, HMGA2 and Dicer, both at RNA and protein levels (Fig 5a, Fig 5b and Fig. 5c).**

HMGA2 is a canonical let-7 targets containing at least six let-7 binding sites (7-mer and 8-mer) in the 3'UTR (Lee et al. (2007) Genes Dev). Dicer also contains three let-7 8-mer sites within its coding sequence (Forman et al (2008) PNAS) and one in the 3'UTR (Tokumaru et al. (2008) Carcinogenesis). This is now clarified in the revised text.

3-2) I wonder whether the 5'isomir of let-7a can include a potential target sites in L1.

This is a very interesting comment. Using RNA22 and RNAhybrid software, **we have found that the 5'isomir let-7a (-1) pairs forming a canonical 7-mer site with the 'bs2rh' binding site (see panel A below)**. In order to determine if the let-7 5'isomir could contribute to control L1 mobilization, we have downloaded the public loci-based *isoform.quantification.txt* files from the TCGA datasets for lung tumor samples used in Fig. 1. To evaluate a possible correlation between somatic L1 retrotransposition in lung cancer and the expression of 5'isomiRs from the different members of let-7 family, we have performed a similar analysis to that carried out in figure 1 (see panel B below). First, we found that the expression was really low (<50 rpm and many with a value of zero). Furthermore, none of them was downregulated in the samples with one or more tumor-specific L1 insertions as we observed for let-7a, let-7e and let-7f. This analysis suggests that, although **the 5'isomiR of let-7 would theoretically bind the L1 mRNA with higher efficiency that let-7, this interaction does not contribute to control L1 mobilization in human lung tumor samples.**

A

	5'	GAG	UCA	G	3'	L1
offset	AGCUG	GCA	C	ACUACCU		
7mer	UUGAU	UGU	G	UGAUGGA		
	3'	A	UG	A	GU	5' Let-7

	5'	GAG	UCA	G	3'	L1
7mer	AGCUG	GCA	C	ACUACCU		
	UUGAU	UGU	G	UGAUGGA		
	3'	A	UG	A	G	5' Iso-let-7

B

Rebuttal Figure 1: A let-7 (-1) isomiR forms a canonical 7mer site with L1 RNA. (A) Predicted pairing, using RNAHybrid, of let-7a and iso-let7a (-1, lacking the U in 5') with L1 bs2. The calculated folding energy is the same in both cases (-20.9 Kcal/mol), however let-7 forms a noncanonical offset 7-mer site while iso-let-7 forms a canonical 7mer site. **(B)** Plot representing the expression levels of let-7 (-1) isomiRs in lung tumor samples without (dark grey, N=14) and with (light grey, N=27) tumor-specific L1 insertions identified by MELT. Differentially expressed miRNAs are marked with * and were identified applying an unpaired two-tailed t test adjusted by FDR<0.01. Expression in reads per million (rpm) is shown. Whiskers were calculated using the Tukey method. Individual black dots represent outliers. Boxes extend from 25th to 75th percentiles, and lines in the middle of the boxes represent the median.

4) Unclear how they normalized the miRNA level in Fig 1b. I wonder a rational why they used the maximum value in each sample to normalize? Each sample could have different maximum level of miRNA actually and forcing to normalize the level of miRNA using the maximum value could include unexpected artifact as well. How the results are different if they use the rank-sum test on the unnormalized level of miRNAs?

We apologize for the confusion generated. We only used the normalization for visualization purposes. **The statistical analysis was performed with the raw rpm values**, both for lung-related miRNAs (supp table III, VII) and for all miRNA expressed >100rpm in lung (supp tables VI,VIII). We have modified the supplementary tables accordingly. Nevertheless, we obtained exactly the same p-values and discoveries with and without normalization, suggesting that normalization does not induce any bias or unexpected artifacts.

Following the reviewer's suggestion, we have performed a Rank Sum test on the unnormalized level of miRNAs and we only find let-7a-2 as significant with an FDR<0.01. If we consider FDR<0.05, then we also get let-7a-1, let-7a-3, and let-7f. This result has been added to our study as Supp. Table IV, and is referred to in the text in page 7: "This correlation was also found for let-7a and le-7f using a different statistical analysis (Rank-sum test, Supplementary Table IV)."

5) As previously reported, newly inserted retrotransposons are often inactivated by epigenetic way. Please the check the L1 RNA level (using RNA-seq) and methylation level in L1 to find the association with the L1 insertion or miRNA levels.

We thank the reviewer for this suggestion. We have now analyzed tumor RNA-seq data from the human lung tumor samples where the tumor-specific somatic L1 insertions were identified (Fig. 1). To determine global TE expression in RNA-seq experiments, we use SQUIRE: Software for Quantifying Interspersed Repeat Elements (Yang et al (2019), Nucleic Acid Res). Interestingly, we have found higher levels of L1Hs mRNAs in the samples with tumor-specific insertions, although the expression of let-7 family members did not correlate with L1Hs RNA levels. This has now been included in of the revised text (page 8):

"Next, we used SQUIRE (Software for Quantifying Interspersed Repeat Elements) (Yang et al. 2019) to quantify L1Hs expression in RNA-seq data from these tumor samples, available in TCGA. **As expected, L1Hs RNA levels were significantly increased in samples with tumor-specific L1 insertions (Supplementary Fig. 1b). However, L1Hs expression negatively correlates with miR-34a but not with let-7 expression (Supplementary Fig. 1b)**"

These data support our original observation, as we have now commented in the discussion, page 25:

"...we have demonstrated that let-7 impairs translation of L1 ORF2p without affecting mRNA stability. This conclusion is further supported by the fact that no correlation between the levels of let-7 and the expression of L1Hs RNA was observed in human lung tumor samples (**Supplementary Fig. 1b**)."

Following reviewer's suggestion, we have investigated a potential correlation between the number of tumor-specific insertions and hypomethylation of L1 promoters. To

predict methylation level of LINE-1 repetitive elements, methylation beta value data was download for each sample from TCGA and analyzed using package REMP (Zheng Y, et al. (2017) Nucleic Acids Res). Unfortunately, only 15 out of 41 samples available data passed the quality controls that REMP package required (Illumina 450k methylation files). When the average methylation of LINE-1 was compared across both sample groups (7 with and 8 without tumor specific L1 insertions samples), there was no evidence of a loss of methylation in samples with *de novo* L1 insertions (see below, Rebuttal Figure 2). We believe that a higher number of samples is necessary to get a reliable result. Nevertheless, we consider that the methylation levels of LINE-1 do not change the main conclusions of our study.

Rebuttal Figure 2. Boxplot of mean DNA methylation level of LINE1 in samples without insertions (blue bar) and with insertions (yellow bar) (8 and 7 samples, respectively).

6) Fig 3d, L1 and HMGA2 enrichment experiments have to be repeated and display the standard deviations of the changes.

We have performed the RNA Immunoprecipitation (RIP) assay three times and consistently observed an enrichment in the amount of L1mRNA bound to AGO2 upon overexpression of let-7 similar to the increase in HMGA2 mRNA. However, the enrichment values vary substantially between experiments, therefore we consider that a representative experiment is more informative than the mean and standard deviation. To show that, we have included a rebuttal figure with a second experiment different to the one represented in Fig. 3d and we have clarified this in the revised manuscript (page 15):

“Strikingly, we observed an enrichment in the amount of L1 mRNA bound to AGO2 upon overexpression of let-7 resembling the behaviour of HMGA2 mRNA (Fig. 3d), a well-known target of let-7”.

Rebuttal Figure 3: An additional representative RNA immunoprecipitation (RIP) of AGO2-Flag in PA-1 cells (embryonic teratocarcinoma cells). Left panel: AGO2-Flag protein was detected by western blot for input and IP. Right panel: mRNA relative enrichment upon let-7 overexpression determined by qRT-PCR.

7) There are more cancers that are involved with genomic instability. I wonder if this let-7 regulation for L1 is also true in other cancer.

This is a very interesting comment. The number of samples available from TCGA with the datasets required for our analysis (WGS from tumor tissue and match healthy solid tissue and tumor sRNA-seq) is reduced and constitutes the limiting factor. In fact, the type of tumor with higher number of samples is lung cancer. However, we have now analysed 36 breast cancer samples in which the number of *de novo* L1 insertions were identified previously by Helman and col. using Transpo-seq (Helman et al. (2014) Genome Research). We did not find any correlation between somatic L1 insertions and miRNA levels. This result has been included in Supplementary Fig 1e, Supplementary Table X and in page 9 of the text:

“Lastly, the same analysis was performed using 36 breast cancer samples which contain a notably smaller number of tumor-specific L1 insertions per sample as determined by Transpo-Seq. No significant correlation was found for any of the 26 miRNAs related to lung cancer (**Supplementary Figure 1e and Supplementary Table X**) suggesting that the contribution of let-7 and mir-34a to L1 mobilization could be specific to some tumor types.”

Reviewer #3 :

In the manuscript titled "The tumor suppressor microRNA let-7 inhibits human LINE-1 retrotransposition" Tristán-Ramos et. al. explored the role of microRNA let-7 in L1 retrotransposition. With well-designed experiments, this study demonstrates that let-7 regulates L1 transposition by inhibiting ORF2p translation and such inhibition is achieved via a potential let-7 binding site within ORF2p, thus providing novel insights into the role of let-7 in maintaining genome integrity. The manuscript is informative and well written, specific comments are listed below.

Figure1 and supplementary figure 1 illustrated the changes in miRNA expression in tumor samples with or without L1 insertions. What are the miRNA expression levels of the matching normal samples? Do patients with L1 insertion also have low let-7

expression in normal samples or such phenomenon can only be observed at the tumor samples?

This is a really interesting question that we asked ourselves during the course of the project. Unfortunately, of the 41 lung tumor samples obtained from TCGA that passed the quality control for our analysis, miRNA-seq data of the matching healthy tissue was only available for 3 of them. As we have now clarified in the text (page 6): “We selected ALL the samples for which whole genome sequencing data from tumor and MATCHED NORMAL LUNG TISSUE, together with tumor miRNA-seq data, were available.” In order to answer the reviewer, we have increased the number of samples by relaxing our original requirements allowing WGS from blood samples as healthy tissue to identify tumor-specific L1 insertions. Nevertheless, the datasets required for the analysis were available only for 7 lung tumor samples from TCGA: TCGA-44-2668, TCGA-49-6742(*), TCGA-50-5930(*), TCGA-50-5932(*), TCGA-43-5670, TCGA-56-7582 y TCGA-90-7767 (*indicates that they had been included in the previous analysis shown in Fig. 1). We identified tumor-specific somatic L1 insertions from WGS data with MELT software (maintaining the same criteria as described in the Methods section): 4 samples contained tumor-specific insertions and 3 did not. We compared the expression levels of the different let-7 family members in normal lung samples with and without insertions. As can be seen in the figure below (as an example we include let-7a-1 and let-7e), we observed a non-significant reduction of let-7a and let-7e levels in the normal tissue that gave rise to tumors with somatic L1 insertions. As control, the expression of let-7 was determined in the matching tumor samples obtaining similar results. Due to the reduced number of samples available in TCGA to perform the analysis, the expression differences were not significant in any case and we cannot draw conclusions about let-7 levels in normal tissue. In our study we have demonstrated that let-7 controls L1 retrotransposition and is clearly downregulated in tumor samples with *de novo* L1 insertions. Our results strongly support the hypothesis that downregulation of let-7 in lung cancer can lead to an increased mobilization of active L1s contributing to tumor progression. Unfortunately, **to answer if a reduction in let-7 levels in healthy tissue can act as a mechanism to allow L1 mobilization causing occasional tumor-initiating insertions requires a number of sequencing data from healthy tissue samples that currently is NOT available.**

Rebuttal Figure 4: Analysis of let-7a-1 and let-7e expression levels in seven lung tumor samples without and with tumor specific L1 insertions (blue bars and yellow bars, respectively) and matching healthy tissues.

The use of both let-7 inhibitors and mimic reasonably demonstrates that let-7 inhibits L1 retrotransposition. However, it would be nice to demonstrate or comment on the specificity of let-7 inhibitors (both within and across let-7 families).

Let-7 family members share not only the seed sequence but a high percentage of the 3' region. This suggests that the let-7a inhibitor binds to the rest of let-7 family members. In fact, it has been shown that let-7a hairpin inhibitor used in this study (Dharmacon) exhibits crossreactivity with other members of the family (Robertson et al (2010) Silence 1;1(1):10). A comment has been included in the page 12 of the revised text: "Although the inhibitor used was designed against let-7a, it has been shown to cross-react with other members of the family (Robertson et al. 2010)".

Page 10 "We were able to recapitulate this effect in a different cell line, HEK293T..", this indicates both let-7 and miR-34a results were recapitulated. What is the endogenous expression level of miR-34a in HEK293T? What's the effect of over-expressing miR-34a in HEK29T? Results need to be included at least as supplementary figures.

We apologize for the lack of clarity. First, we have now included the levels of miR-34a in HEK293T cells, that are slightly higher than in HeLa (see updated Supplementary Fig. 2a). Second, we have also included a retrotransposition assay overexpressing miR-34 in HEK293T cells (see Supplementary Fig. 2d). This has now been commented on the revised text (pg. 12):

"Conversely, miR-34 overexpression in HEK293T cells, where the endogenous levels are slightly higher than in HeLa (**Supplementary Fig. 2a**), led to an increase in L1 retrotransposition using the dual luciferase reporter vector pYX014 (**Supplementary Fig. 2d**). The different effects observed for miR-34 overexpression in HeLa (**Fig. 2b**) and HEK293T cells (**Supplementary Fig. 2d**) suggest a potential indirect, cell-type specific effect of miR-34 on L1 mobilization."

We have further commented our mir-34 results in discussion (page 24): "It is worth noting that the expression of another tumor suppressor miRNA, miR-34a, is also reduced in lung tumors with L1 activity and correlates negatively with L1Hs RNA levels. However, we did not observe a consistent effect of the latter on L1 retrotransposition, under our experimental conditions. We speculate that mir-34 could indirectly regulate L1 mobilization, targeting a member of the epigenetic regulatory network controlling expression of active L1s in our genome."

Page 12 "we performed the luciferase-based retrotransposition assay in two lung cancer cell lines with significantly different endogenous levels of let-7... depletion of let-7 increased L1 retrotransposition by 2.5 times on average (Fig. 2d)" Do the two cell line exhibit different L1 retrotransposition activity under control condition? And in general, does endogenous let-7 expression correlate with L1 retrotransposition activity?

Indeed, both cell lines exhibit different retrotransposition activity under control condition. A549 cells support higher levels of engineered L1 retrotransposition despite having higher levels of let-7 microRNA (See Rebuttal Figure 5). This result is not against our main conclusion that let-7 controls L1 retrotransposition. Multiple mechanisms of control, both at transcriptional and post-transcriptional levels, act on somatic cells as a defence against L1 insertions (Goodier (2016) Mob DNA). Given the different genetic background of both tumor cell types, beyond microRNAs, alteration of diverse mechanisms involved in L1 mobility is expected.

Rebuttal Figure 5. Relative endogenous let-7a levels (blue bars) and engineered L1 retrotransposition (yellow bars) in A549 and SK-MES-1 lung tumor cell lines.

Regarding the second question, most of the mature let-7 family members are undetectable or little expressed in human and mouse embryonic early development stage, and the level of let-7 increases upon differentiation (Pasquinelli et al. (2000) Nature). By contrast, L1 mobilization occurs early in human and mouse embryogenesis and are silent in most of differentiated cells (Schumann et al. (2019) Mobile DNA). To know if this observation involves a causal correlation would require further investigation and we feel that this is beyond the timeline/scope of this study. Please, see also response to point 7 from reviewer 2.

Figure 5e, quantification needed (something similar as performed in 5d)

We used the ZenBlue software to quantify the double positive cells, however we had not included this because of the low sample size. Given the low transfection efficiency of pVan583, and the highly inefficient translation of ORF2p-mCherry, only around 0.3% of U2OS cells are GFP+ and mCherry+ (1.5% are GFP+ and from those only 18% are mCherry+), therefore we did not detect enough number of cells to be quantified using this method (see below, rebuttal figure 6). That is the reason why we turned to flow cytometry, a more sensitive and quantitative approach: 10^5 cells were passed through the cytometer getting at least 3500 GFP positive cells, allowing us to get more robust results to sustain our conclusions. We have now included more details about the quantitative analysis in the methods section of revised text (pg. 41) and a representative FACS histogram for each condition in Fig 5d. Furthermore, we have moved the microscopy images to Supp Fig 5q and enlarged them, as reviewer #1 also requested.

Rebuttal Figure 6. Quantification of fluorescent intensity in mCherry+ and GFP+ cells by confocal microscopy in U2OS cells transfected with pVan583 under scr (N=3) and let-7 mimic (N=5) conditions.

We have also updated the references including now the papers mentioned above and the following ones:

- Rodriguez-Martin, B. et al. Pan-cancer analysis of whole genomes identifies driver rearrangements promoted by LINE-1 retrotransposition. *Nat Genet* (2020).
- Tristan-Ramos, P. et al. sRNA/L1 retrotransposition: using siRNAs and miRNAs to expand the applications of the cell culture-based LINE-1 retrotransposition assay. *Philos Trans R Soc Lond B Biol Sci* **375**, 20190346 (2020).
- Flasch, D.A. et al. Genome-wide de novo L1 retrotransposition connects endonuclease activity with replication. **177**, 837-851 e28 (2019).
- Sultana, T. et al. The Landscape of L1 Retrotransposons in the Human Genome Is Shaped by Pre-insertion Sequence Biases and Post-insertion Selection. *Mol Cell* **74**, 555-570 e7 (2019).
- Monot, C. et al. The specificity and flexibility of L1 reverse transcription priming at imperfect T-tracts. *PLoS Genet* **9**, e1003499 (2013).
- Lian, S.L. et al. The C-terminal half of human Ago2 binds to multiple GW-rich regions of GW182 and requires GW182 to mediate silencing. *RNA* **15**, 804-13 (2009).

REVIEWER COMMENTS

Reviewer #1 (Remarks to the Author):

(1) Regarding grammar and supplement reorganization  OK

(2) Regarding L1 RNP description  OK

(3) Regarding supplement contents  problems, see below

a. Some of these blot have extra lanes on them (unlabeled) - I presume that is because the data shown in those lanes is not relevant to the experiments / science presented but it does interfere with 'at a glance' interpretation. I do not insist the authors crop these images at this point (although, they could do it).

b. IP Supp Fig 3c  there is apparently ORF1p signal in the mock IP? Why? Also, what is going on with the massive background on this blot? The signal for the target protein should not be the weakest signal on the blot. Also, why is the background for AGO and ORF1p antibodies identical? Was the blot not stripped between probings? The ORF1p signal pointed to here is at ~37 kDa, yet on the blot above (Input Supp Fig 3c) it is at below 37 kDa, and the blot below it (Fig 5b -- also, massive background?) is above 37 kDa; so which is it? The migration differences are small, but I have run 100,000 western blots and, using the same marker, running buffer, and run time, you should not usually see this kind of variation. This data has made me skeptical that the signal labeled as ORF1p is actually ORF1p .. in particular in the IP Supp Fig 3c blot which is important because the authors are claiming that the ORF1p interactions is RNA-dependent based on a decrease in ORF1p level after adding RNase -- the + RNase samples do indeed resemble the signal strength of the mock IP, but it also strikes me as exceptionally unusual that the RNase treatment should render the residual binding to precisely the level of the mock - presumably the ORF1p in the mock is also an RNP (since it is not RNase treated) thus, RNase treatment should also affect this sample, reducing its baseline further? As it stands I am shaky on this data and interpretation. Assuming this is ORF1p signal, an alternative interpretation is that ALL the ORF1p is non-specific (whether mock or with AGO1/2, although elevated in AGO). Being ORF1p is an RNA binding protein it would not be unusual for post-lysis association with other RNA-bound RBPs to occur, and since the interactions is (at least partly) through RNA, treatment with RNase should nearly always reduce its level to one degree or another. A better control is needed to discriminate these possibilities. It's also not clear to me how crucial this claim is for the paper, which can be pinned upon the transcript levels and insertional activity and does not need to show association of ORF1p (only, potentially, to determine if post-translational L1 RNPs are also among targets). To this point are the authors certain that signal in the 293T input (Input Supp Fig 3c) is ORF1p? To our experience some enrichment either by sucrose cushion or IP would normally be required to obtain this kind of signal. Some additional controls might be useful here - an alternative validated antibody (like the 4H1 monoclonal), other previously validated

different expressing cell lines (see e.g. Cristofari reference that follows). The author's say protein concentration was measured in the methods but not how many micrograms were loaded. In Philippe, C. et al (2016). Activation of individual L1 retrotransposon instances is restricted to cell-type dependent permissive loci. eLife 5(MARCH2016), 166. -- low-level ORF1p signal (with extremely low background) is shown from 293T with 10ug loaded but using a different polyclonal. Migration is above 40kDa, but running system probably different. Take home message: these ORF1p blots are not high enough quality to stand with the claims that ORF1p is present and RNase sensitive in these IPs, and the background in some of these blots anyway, leaves plenty to be desired in terms of data quality / scrutiny. Lastly, the result seems to indicate that the amount of ORF1p in the whole cell extract is plenty for ready detection, yet, when apparently co-enriched with Ago there's less signal? That's not enrichment at all. The authors should show the degree of immuno-depletion of Ago and ORF1p (Input and flow through from IP) on the same blot as the elution (with % loaded). This would demonstrate the degree of capture and the relative enrichment of the targets. Otherwise, I suggest removing the claims made from these data entirely. Sorry, I would have liked to give this feedback in the last round of reviewing but I could not interpret the supplement - underscoring the importance of my original critique. In my opinion this experiment would better be done in PA-1 cells - it would better accord to the rationale of Fig. 3D using an accepted model for endogenous expression and L1 insertional activity. Which leads me to my next confusion.

Strangely, the authors write (in methods): "Co-immunoprecipitation
736 PA-1 cells were transfected with FLAG-AGO1/2, or an empty plasmid ('mock').
737 48h post transfection, cells were lysed using lysis buffer and
738 immunoprecipitation with anti-FLAG M2 mouse (Sigma, F3165) as it was
739 described above. After the last wash with lysis buffer, the beads were treated
740 with 100 µg/ml RNase A for 30 min. The western blot was performed with Anti-
741 ORF1p (1:1000) provided by Dr. Oliver Weichenrieder, (Max-Planck, Germany)
742 and subsequently, with anti-FLAG M2 mouse (Sigma, F3165).

But, where is this anti-ORF1p blot? It should CLEARLY be stated which figures this applies to!

In main text this is written as: Interestingly, we observed by co-
297 immunoprecipitation in HEK 293T cells that overexpressed FLAG-tagged
298 AGO2 and AGO1 proteins interact with endogenous L1-ORF1p in an RNA-
299 dependent manner (Supplementary Fig. 3c).

My critique of the blot quality holds, regardless - but, now I am confused, still, which cell line was used where. This needs to be addressed.

(4) Regarding Mock IP definition  OK

(5) Regarding dual-tagged experiments  Fig. 5e - although apparently statistically significant, the

effect size appears to be minuscule, that coupled with the low transposition rate (~1/3rd vs untagged - a much larger effect size than the let-7 inhibition imparts on the tagged construct - if I am understanding correctly), and forced expression context make me skeptical of the additional value / biological significance of this particular data/interpretation. Nevertheless, the authors have offered full disclosure of experimental details in the revisions and this is something I think expert readers can evaluate for themselves given all the information provided. OK.

(6) Regarding Figure 6  OK

(7) Regarding cell line usage  looks like more care is needed - see my comments above.

Signed,
John LaCava

Reviewer #2 (Remarks to the Author):

All initially concerns raised by me were properly addressed by extensively additional analyses and experiments. Particularly, finding a shifted 7-mer site (bs2rh) in L1 ORF and reporter assays with the site clearly showed the L1 ORF translational regulation by let-7. However, tumor-specific relationship b/w the L1 insertion and let-7 expression raises another question why the interaction is working in specific tumors but not in others. They need to discuss this point in the discussion.

Reviewer #3 (Remarks to the Author):

The authors have addressed all the points carefully. Thank you for providing such a detailed and thorough rebuttal letter during this special time!

POINT BY POINT RESPONSE TO REVIEWERS

Reviewer #1

(1) Regarding grammar and supplement reorganization  OK

We are pleased to have addressed this reviewer's concern.

(2) Regarding L1 RNP description  OK

We are pleased to have addressed this reviewer's concern.

(3) Regarding supplement contents  problems, see below

a. Some of these blot have extra lanes on them (unlabeled) - I presume that is because the data shown in those lanes is not relevant to the experiments / science presented but it does interfere with 'at a glance' interpretation. I do not insist the authors crop these images at this point (although, they could do it).

To clarify the figure while keeping the uncropped versions of blots, as encouraged by Nature Communications policy, in the revised version of the manuscript we have now framed the parts of the blots that are actually used in the main figures. We have also labelled all the lanes and bands that could help interpreting the results such as bands corresponding to the IgG chains and lanes corresponding to untransfected samples. Furthermore, we have included molecular weight markers for all blots. They were cut and pasted from the same image obtained by chemiluminescence, but adjusting the contrast to visualize it directly when possible, or from the image of the blot obtained by epi-illumination for prestained markers. We hope that this updated presentation will be a good balance between reporting raw data and providing 'at a glance' interpretation.

b. IP Supp Fig 3c  there is apparently ORF1p signal in the mock IP? Why? Also, what is going on with the massive background on this blot? The signal for the target protein should not be the weakest signal on the blot. Also, why is the background for AGO and ORF1p antibodies identical? Was the blot not stripped between probings? The ORF1p signal pointed to here is at ~37 kDa, yet on the blot above (Input Supp Fig 3c) it is at below 37 kDa, and the blot below it (Fig 5b -- also, massive background?) is above 37 kDa; so which is it? The migration differences are small, but I have run 100,000 western blots and, using the same marker, running buffer, and run time, you should not usually see this kind of variation. This data has made me skeptical that the signal labeled as ORF1p is actually ORF1p . in particular in the IP Supp Fig 3c blot which is important because the authors are claiming that the ORF1p interactions is RNA-dependent based on a decrease in ORF1p level after adding RNase -- the + RNase samples do indeed resemble the signal strength of the mock IP, but it also strikes me as exceptionally unusual that the RNase treatment should render the residual binding to precisely the level of the mock - presumably the ORF1p in the mock is also an RNP (since it is not RNase treated) thus, RNase treatment should also affect this sample, reducing its baseline further? As it stands I am shaky on this data and interpretation. Assuming this is ORF1p signal, an alternative interpretation is that ALL the ORF1p is non-specific (whether mock or with AGO1/2, although elevated in AGO). Being ORF1p is an RNA binding protein it would not be unusual for post-lysis association with other RNA-bound RBPs to occur, and since the interactions is (at least partly) through RNA, treatment with RNase should nearly always reduce its level to one degree or another. A better control is needed to discriminate these possibilities. It's also not clear to me how crucial this claim is for the paper, which can be pinned

upon the transcript levels and insertional activity and does not need to show association of ORF1p (only, potentially, to determine if post-translational L1 RNPs are also among targets). To this point are the authors certain that signal in the 293T input (Input Supp Fig 3c) is ORF1p? To our experience some enrichment either by sucrose cushion or IP would normally be required to obtain this kind of signal. Some additional controls might be useful here - an alternative validated antibody (like the 4H1 monoclonal), other previously validated different expressing cell lines (see e.g. Cristofari reference that follows). The author's say protein concentration was measured in the methods but not how many micrograms were loaded. In Philippe, C. et al (2016). Activation of individual L1 retrotransposon instances is restricted to cell-type dependent permissive loci. eLife 5(MARCH2016), 166. -- low-level ORF1p signal (with extremely low background) is shown from 293T with 10ug loaded but using a different polyclonal. Migration is above 40kDa, but running system probably different. Take home message: these ORF1p blots are not high enough quality to stand with the claims that ORF1p is present and RNase sensitive in these IPs, and the background in some of these blots anyway, leaves plenty to be desired in terms of data quality / scrutiny. Lastly, the result seems to indicate that the amount of ORF1p in the whole cell extract is plenty for ready detection, yet, when apparently co-enriched with Ago there's less signal? That's not enrichment at all. The authors should show the degree of immuno-depletion of Ago and ORF1p (Input and flow through from IP) on the same blot as the elution (with % loaded). This would demonstrate the degree of capture and the relative enrichment of the targets. Otherwise, I suggest removing the claims made from these data entirely. Sorry, I would have liked to give this feedback in the last round of reviewing but I could not interpret the supplement - underscoring the importance of my original critique. In my opinion this experiment would better be done in PA-1 cells - it would better accord to the rationale of Fig. 3D using an accepted model for endogenous expression and L1 insertional activity. Which leads me to my next confusion.

Strangely, the authors write (in methods): "Co-immunoprecipitation
736 PA-1 cells were transfected with FLAG-AGO1/2, or an empty plasmid ('mock').
737 48h post transfection, cells were lysed using lysis buffer and
738 immunoprecipitation with anti-FLAG M2 mouse (Sigma, F3165) as it was
739 described above. After the last wash with lysis buffer, the beads were treated
740 with 100 µg/ml RNase A for 30 min. The western blot was performed with Anti-
741 ORF1p (1:1000) provided by Dr. Oliver Weichenrieder, (Max-Planck, Germany)
742 and subsequently, with anti-FLAG M2 mouse (Sigma, F3165).

But, where is this anti-ORF1p blot? It should CLEARLY be stated which figures this applies to!

In main text this is written as: Interestingly, we observed by co-
297 immunoprecipitation in HEK 293T cells that overexpressed FLAG-tagged
298 AGO2 and AGO1 proteins interact with endogenous L1-ORF1p in an RNA-
299 dependent manner (Supplementary Fig. 3c).

My critique of the blot quality holds, regardless - but, now I am confused, still, which cell line was used where. This needs to be addressed.

We thank the reviewer for raising this concern. Considering it, as well as the reviewer’s reasoning about this experiment not being indispensable to support the conclusions of the study (“It’s also not clear to me how crucial this claim for the paper, which can be pinned upon the transcript levels and insertional activity and does not need to show association of ORF1p”), we have decided to remove these data entirely. We are convinced that the findings of this work remain robustly supported, and we apologize for the confusion generated by this experiment.

Even having decided to remove this experiment, we would like to address the reviewer’s concern about this experiment (now rebuttal figure 1) ‘point-by-point’.

Rebuttal Figure 1. Co-immunoprecipitation (co-IP) of FLAG-AGO1/2 and endogenous ORF1p in the presence/absence of RNase.

i. *there is apparently ORF1p signal in the mock IP? Why?*

Indeed, there is ORF1p signal in the mock IP which is likely due to unspecific association between this protein and the beads or antibody. These unspecific associations are often more prominent when elution is performed by heating the beads – as done here – compared to a more specific approach using an excess of Flag-peptide. However, this signal is much less intense than the test IP (compare lane 1 with 2 and 4 in Rebuttal Figure 1, blot d).

ii. *Also, what is going on with the massive background on this blot? The signal for the target protein should not be the weakest signal on the blot. Also, why is the background for AGO and ORF1p antibodies identical? Was the blot not stripped between probings?*

Note that the target protein of this IP was Ago-Flag (and not ORF1p, only a fraction of which was co-immunoprecipitated with Ago-Flag). Regarding the additional bands observed on these blots, the immunoprecipitation of Flag-AGO was performed with a mouse anti-FLAG M2 and the elution was carried out by heating the samples at 70°C for 20 min (This also applies to IP in fig. 3, as it has now been further clarified in Methods). Then, the same primary antibody was used for immunoblot detection of Flag-AGO. As a consequence, the anti-mouse secondary antibody recognized the Flag-Ago (~100 kDa, top darkest band of the

membrane, Rebuttal Figure, blot c), as well as the denatured heavy and light chains of the anti-Flag M2 antibody used for the IP (~50 and 25 kDa, respectively). This explains most of the extra-bands detected in the blot. We have now labelled these extra-bands in Supplementary Figure 6 and rebuttal Fig 1. After detection of Flag-AGO, the IP membrane was directly incubated with the anti-hORF1 antibody (Rebuttal figure 1, blot d). As deduced by this reviewer, the blot was indeed not stripped between probings. We did not find it necessary since AGO2 and ORF1p have markedly different sizes. Thus, the background after detection of hORF1p included all the bands previously described (Rebuttal figure 1, blot d). We believe that this simplified procedure did not interfere with the interpretation of the experiment.

- iii. *The ORF1p signal pointed to here is at ~37 kDa, yet on the blot above (Input Supp Fig 3c) it is at below 37 kDa, and the blot below it (Fig 5b -- also, massive background?) is above 37 kDa; so which is it? The migration differences are small, but I have run 100,000 western blots and, using the same marker, running buffer, and run time, you should not usually see this kind of variation. This data has made me skeptical that the signal labeled as ORF1p is actually ORF1p*

We have also used this antibody numerous times, thus we are completely sure that the labelled band corresponds to ORF1p in all cases. The subtle differences in size are reflecting imprecise alignment of the marker lines in the figure, rather than actual differences in the size of the band. We have now included the original bands of the marker to ensure that they are precisely placed now (Rebuttal figure 1).

- iv. *...it also strikes me as exceptionally unusual that the RNase treatment should render the residual binding to precisely the level of the mock – As it stands I am shaky on this data and interpretation. Assuming this is ORF1p signal, an alternative interpretation is that ALL the ORF1p is non-specific (whether mock or with AGO1/2, although elevated in AGO). Being ORF1p is an RNA binding protein it would not be unusual for post-lysis association with other RNA-bound RBPs to occur, and since the interactions is (at least partly) through RNA, treatment with RNase should nearly always reduce its level to one degree or another. A better control is needed to discriminate these possibilities. It's also not clear to me how crucial this claim is for the paper, which can be pinned upon the transcript levels and insertional activity and does not need to show association of ORF1p (only, potentially, to determine if post-translational L1 RNPs are also among targets).*

We believe that the mock signal represents unspecific binding of ORF1p directly to the Dynabeads protein G or to the constant region of the antibody, therefore we consider plausible that RNase treatment removes the interaction between the L1-RNP and AGO and leaves in the IP only the amount of ORF1p that was bound non-specifically to the beads or antibody. However, we agree with this reviewer that we cannot exclude that this interaction occurred post-lysis through an RNA bridge. Thus, given that this experiment is not central to the manuscript, and following the reviewer recommendation, we decided to remove this figure from the manuscript.

- v. *To this point are the authors certain that signal in the 293T input (Input Supp Fig 3c) is ORF1p? ... In Philippe, C. et al (2016). Activation of individual L1 retrotransposon instances is restricted to cell-type dependent permissive loci. eLife 5(MARCH2016), 166. -- low-level ORF1p signal (with extremely low background) is shown from 293T with 10ug*

loaded but using a different polyclonal. Migration is above 40kDa, but running system probably different.

As stated above, we have performed this type of western-blot numerous times and are convinced that the labelled band corresponds to ORF1p. Regarding the reference brought up by the reviewer, it is difficult to directly compare the signal intensity of western-blots, since different antibodies and detection methods (fluorescent vs ECL) were used. We loaded 40 µg of protein in our gels and ORF1p was readily detected with extremely low background as well (please, have a look to blot b in Rebuttal Figure 1). In this case, the input membrane was first incubated with the anti-hORF1 antibody (Rebuttal Figure 1, blot b) and afterwards, with the anti-flag antibody (Rebuttal Figure 1, blot a).

- vi. Take home message: these ORF1p blots are not high enough quality to stand with the claims that ORF1p is present and RNase sensitive in these IPs, and the background in some of these blots anyway, leaves plenty to be desired in terms of data quality / scrutiny. (...) Otherwise, I suggest removing the claims made from these data entirely.

Although we hope to have addressed the reviewer's concern about this figure, and in order to avoid any misinterpretation from the readers, we have removed the entire Supplementary Fig. 3c from the manuscript following reviewer's recommendation.

- vii. Strangely, the authors write (in methods): "Co-immunoprecipitation: PA-1 cells were transfected...."

We apologize to the reviewer for the confusion regarding the cell line used in the co-immunoprecipitation experiments. This was a mistake during manuscript preparation. For the sake of precision, we would like to clarify that the experiment was performed in HEK293T cells, as was stated in the main text. Nevertheless, this will not cause further trouble as, following reviewer's advice, we have removed the experiment in Supp Fig. 3c.

(4) Regarding Mock IP definition  OK

We are glad to have addressed this reviewer's concern.

(5) Regarding dual-tagged experiments  Fig. 5e - although apparently statistically significant, the effect size appears to be minuscule, that coupled with the low transposition rate (~1/3rd vs untagged - a much larger effect size than the let-7 inhibition imparts on the tagged construct - if I am understanding correctly), and forced expression context make me skeptical of the additional value / biological significance of this particular data/interpretation. Nevertheless, the authors have offered full disclosure of experimental details in the revisions and this is something I think expert readers can evaluate for themselves given all the information provided. OK.

We thank the reviewer for this comment. Indeed, we agree that we have thoroughly provided details and information so that any reader can understand the limitations of this particular experiment. However, we believe this experiment does in fact provide additional value to the experiments involving the tagged ORF2p construct (in Fig. 5c and Supp Fig. 5). In pVan583, ORF2p is translated from the bicistronic L1 RNA by the unconventional termination/re-initiation mechanism (Alisch et al., Genes & Dev 2006) that governs endogenous ORF2p expression, whereas in pSA500, ORF2 is robustly expressed from a CMV promoter and a monocistronic

transcript.

(6) Regarding Figure 6  OK

We are glad that the figure is now more clear and easier to follow.

(7) Regarding cell line usage  looks like more care is needed - see my comments above.

Please, see response 3.b.vii above regarding cell line usage.

Reviewer #2 (Remarks to the Author):

All initially concerns raised by me were properly addressed by extensively additional analyses and experiments. Particularly, finding a shifted 7-mer site (bs2rh) in L1 ORF and reporter assays with the site clearly showed the L1 ORF translational regulation by let-7. However, tumor-specific relationship b/w the L1 insertion and let-7 expression raises another question why the interaction is working in specific tumors but not in others. They need to discuss this point in the discussion.

We thank the reviewer for the comments and suggestions provided in the last revision. We believe they have enabled us to improve our study with more robust data. Regarding the second point accurately brought up by the reviewer, we have now included in the discussion section (pag. 26 line 552):

“Notably, even though the let-7 miRNA-L1 mRNA interaction likely occurs in any cell that simultaneously expresses both RNAs, we did not observe any correlation between increased somatic L1 insertions and reduced let-7 levels in human breast cancer samples (**Supplementary Fig. 1e**). We speculate that in some cell types, other regulatory layers that suppress L1 mobilization at transcriptional or post-transcriptional levels (Pizarro, et al. 2016) may overshadow miRNA-mediated L1 inhibition. Consistently, L1 reactivation is less frequent in breast cancer than in other tumor types (Helman, et al. 2014; Tubio, et al. 2014, Rodríguez-Martin et al. 2020), suggesting that in breast cancer an additional mechanism of control could be restricting L1 insertions despite a reduced expression of let-7”.

Reviewer #3 (Remarks to the Author):

The authors have addressed all the points carefully. Thank you for providing such a detailed and thorough rebuttal letter during this special time!

We thank the reviewer for the previous suggestions and comments that have improved our study, and for the generous appreciation. Indeed, it has not been easy to work during this time but we believe that only science will allow us to overcome this situation.

REVIEWERS' COMMENTS:

Reviewer #1 (Remarks to the Author):

The authors have satisfactorily responded to my critiques - I have no further comments at this time and approve publication.